# Decomposition of Equivariant Maps via Invariant Maps: Application to Universal Approximation under Symmetry

**Akiyoshi Sannai**[*][†]                                                                  *sannai.akiyoshi.7z@kyoto-u.ac.jp*
*Department of Physics, Kyoto University, RIKEN*
*Kitashirakawa, Sakyo, 606-8502, Kyoto, Japan*

**Yuuki Takai**[*]                                                                          *takai@neptune.kanazawa-it.ac.jp*
*Kanazawa Institute of Technology*
*7-1 Ohgigaoka, Nonoichi, 921-8501, Ishikawa, Japan*

**Matthieu Cordonnier**                                                                *matthieu.cordonnier@gipsa-lab.fr*
*GIPSA-lab Grenoble INP, Grenoble INP, Université Grenoble Alpes*
*11 rue des Mathématiques, Grenoble Campus BP46, F-38402 SAINT MARTIN D'HERES CEDEX, France*

**Reviewed on OpenReview:** *https://openreview.net/forum?id=ycOLyHh1Ue*

## Abstract

In this paper, we develop a theory about the relationship between invariant and equivariant maps with regard to a group $G$. We then leverage this theory in the context of deep neural networks with group symmetries in order to obtain novel insight into their mechanisms. More precisely, we establish a one-to-one relationship between equivariant maps and certain invariant maps. This allows us to reduce arguments for equivariant maps to those for invariant maps and vice versa. As an application, we propose a construction of universal equivariant architectures built from universal invariant networks. We, in turn, explain how the universal architectures arising from our construction differ from standard equivariant architectures known to be universal. Furthermore, we explore the complexity, in terms of the number of free parameters, of our models, and discuss the relation between invariant and equivariant networks' complexity. Finally, we also give an approximation rate for $G$-equivariant deep neural networks with ReLU activation functions for finite group $G$.

## 1 Introduction

Symmetries play a fundamental role in many machine learning tasks. Incorporating these symmetries into deep learning models has proven to be a successful strategy in various contexts. Notable examples include the use of convolutional neural networks (CNNs) to address translation symmetries (LeCun et al., 2015; Cohen & Welling, 2016), deep sets and graph neural networks for permutation symmetries (Zaheer et al., 2017; Scarselli et al., 2008; Kipf & Welling, 2017; Defferrard et al., 2016), and spherical CNNs for rotation symmetries (Cohen et al., 2018; Esteves et al., 2020). The underlying common principle in all these cases is as follows: once the inherent symmetries of a target task are identified, the learning model is designed to encode these symmetries. In doing so, we aim to improve the quality of learning by building a model that fits the characteristics of the task better.

In mathematics, symmetries are represented by the concepts of groups and group actions. A group is a set of transformations, and the action of a group results in a transformation of a given set. The symmetries of group actions are usually divided into two categories: *invariant* tasks and *equivariant* tasks. *Invariant*

---

[*]Both authors contributed equally to this research.
[†]This work includes results obtained while the author was affiliated with the Matsuo-Iwasawa Laboratory at The University of Tokyo, Japan.

tasks require the output to remain unchanged by any transformation of the input via the group action. On the other hand, in *equivariant* tasks, a transformation of the input results in a similar transformation of the output. For example, in computer vision, object detection is an invariant task whereas image segmentation is an equivariant task with regard to rotations and shift transformations.

Despite appearing in tasks of different natures, there are some mathematical relations between invariant and equivariant maps. One can easily verify that the composition of an equivariant map followed by an invariant one results in an invariant map. This relation is at the core of convolutional architectures, which consist of layers made of an equivariant convolution followed by invariant pooling (LeCun et al., 2015). The same is done by Maron et al. (2019b; 2020) to define $G$-invariant networks for a finite group $G$.

In this paper, we explore the relation between invariant and equivariant maps. Our main result Theorem 1 states that for a given group $G$ acting on a set, there is a *one-to-one correspondence* between $G$-equivariant maps and $H_i$-invariant maps, where the $H_i$ are some stabilizer subgroups of $G$. This allows us to reduce any equivariant tasks to some invariant tasks and vice versa.

As a main application of Theorem 1, we study *universal approximation* for equivariant maps. The *universal approximation theorem* (Pinkus, 1999), which is fundamental in deep learning, asserts that any reasonably smooth function can be approximated by an artificial neural network with arbitrary accuracy. In the presence of symmetries, we enforce the neural network architecture to match the symmetries of the target tasks, and, by doing so, we significantly reduce the hypothesis class of neural networks. Naturally, it is essential to ensure that the universal approximation property is not compromised.

It turns out that the universal approximation problem is more involved in the equivariant than in the invariant setup. Indeed, for the symmetric group of permutations of $n$ elements $S_n$, Zaheer et al. (2017) showed that the *DeepSets* invariant architecture is universal via a representation theorem which is famous as a solution for Hilbert's 13th problem by Kolmogorov (1956) and Arnold (1957). However, they do not provide such a theoretical guarantee for their equivariant architecture. Their result for invariance was then extended to more general groups by Maron et al. (2019b) and Yarotsky (2022), but it is another series of papers which later solved the equivariant case, each with their own techniques (Keriven & Peyré, 2019; Segol & Lipman, 2019; Ravanbakhsh, 2020).

Using our decomposition Theorem 1, we propose, in Theorem 2 an alternative way to build universal $G$-equivariant architectures via $H_i$-invariant maps for some suitable subgroups $H_i \subset G$. As we explain in Remarks 1 and 2, the equivariant universal approximator that we obtain is different from others in the literature.

As a second application of our main theorem, we examine the number of parameters as well as the rate of approximation by invariant/equivariant neural networks with ReLU activation. In Theorem 3, we provide both lower and upper bounds on the minimal number of parameters required to approximate an equivariant map to a given accuracy with regard to the minimal number of parameters required to approximate an invariant map in that same accuracy. Finally, in Theorem 4 and Corollary 1, we give an approximation rate for $G$-equivariant neural networks among $G$-equivariant functions with Hölder smoothness condition. This last result is an extension of a result from Sannai et al. (2021) from $S_n$ to any finite group $G$.

## 1.1 Contributions

Our contributions are summarized as follows:

- We introduce a relation between invariant maps and equivariant maps. This allows us to reduce some problems for equivariant maps to those for invariant maps.

- We apply the relation to constructing universal approximators by equivariant deep neural network models for equivariant maps. Our models for universal approximators are different from the standard models, such as the ones from Zaheer et al. (2017) or Maron et al. (2019a). However, the number of parameters in our models can be very small compared to the fully connected models.

- As other applications of the relation, we show some inequalities among invariant and equivariant deep neural networks regarding their ability to approximate any invariant and equivariant continuous maps, respectively, for a given accuracy. Moreover, we show an approximation rate of $G$-invariant and $G$-equivariant ReLU deep neural networks for elements of a Hölder space.

## 1.2 Related work

**Symmetries in machine learning**   Symmetries have been considered since the early days of machine learning by Shawe-Taylor (1989; 1993); Wood & Shawe-Taylor (1996). In contemporary deep learning, they keep generating increasing interest following seminal works from Kondor (2008); Gens & Domingos (2014); Qi et al. (2017); Ravanbakhsh et al. (2017); Zaheer et al. (2017). The most commonly encountered symmetries involve translations, addressed by convolutional architectures (LeCun et al., 2015) and their generalization to arbitrary compact groups (Cohen & Welling, 2016; Kondor, 2008); rotations (Cohen et al., 2018; Esteves et al., 2020); as well as permutations. The latter are particularly relevant for set data (Zaheer et al., 2017; Qi et al., 2017; Maron et al., 2020) as well as for graph data via most of the graph neural network architectures (Scarselli et al., 2008; Defferrard et al., 2016; Kipf & Welling, 2017; Bruna et al., 2014). Permutation symmetries are also present in modern Transformers of architectures, via self attention (Vaswani et al., 2017) and positional encoding. Recent extension of Transformers for graph learning as proposed by Kim et al. (2021), utilize Laplacian eigenvectors for positional encoding features. Thus, to address ambiguity in eigenvector choices, Lim et al. (2023) recently proposed an architecture that is invariant to change of basis in the eigenspaces. Villar et al. (2021; 2024) are interested in the symmetries arousing from classical physics and the distinction between "passive symmetries" coming from physical law and being empirically observed, and "passive symmetries" coming from arbitrary choice of design such as labeling of elements of a set or nodes of a graph. Regarding the connection between invariant and equivariant maps, this work is a direct follow-up of Sannai et al. (2019). The methods and results from Theorems 1 and 2 extend the content of Sannai et al. (2019) from the symmetric group $S_n$ to an arbitrary group. More recently, Blum-Smith & Villar (2023) explain how to build equivariant maps from invariant polynomials using invariant theory. Finally, we shall mention that invariant/equivariant deep learning is now a fast growing field with a plethora of various architectures. Some researchers have recently proposed to unify most of existing approaches through a general framework known as Geometric Deep Learning (Bronstein et al., 2021).

**Symmetries and universal approximation**   Universal Approximation property is fundamental classical deep learning, extensively studied since the pioneer works by Cybenko (1989); Hornik et al. (1989); Funahashi (1989); Barron (1994); Kůrková (1992); Pinkus (1999) and more and further explored by Sonoda & Murata (2017); Hanin & Sellke (2017); Hanin (2019). Concerning invariant architecture, Yarotsky (2022) observed that it is straightforward to build a universal invariant architecture from universal classic architecture just by group averaging when the group is finite. However, this is unfeasible for large groups such as $S_n$. Additionally, this issue was recently addressed by Sannai et al. (2024) who showed that, it can sometimes be enough to average over a subset of the group rather that the entire group. By doing so, they are, for instance, able to reduce the averaging operation complexity from $O(n!)$ to $O(n^2)$ in the case of graph neural networks. Zaheer et al. (2017) prove universality of DeepSets via a representation theorem from Kolmogorov (1956) and Arnold (1957) coupled with a sum decomposition. The effectiveness of this universal decomposition for continuous functions on sets is discussed by Wagstaff et al. (2022), who argue that the underlying latent dimension must be high-dimensional. Recently, Tabaghi & Wang (2023) improved the universality result from Zaheer et al. (2017). Some generalizations to other groups have been proposed by Maron et al. (2019b) and Yarotsky (2022), and universality of some invariant graph neural networks was proved by Maron et al. (2019b) and Keriven & Peyré (2019).

In the case of equivariant networks, Keriven & Peyré (2019); Segol & Lipman (2019); Ravanbakhsh (2020) established universality results for finite groups using high order tensors. More recently, Dym & Maron (2021) suggested a universal architecture for rotation equivariance and Yarotsky (2022) for the semi-direct product of $\mathbb{R}^n$ and $SO_n(\mathbb{R})$ (translation plus rotation). In this paper, we propose circumventing the challenge of directly addressing equivariant universality by constructing equivariant universal architectures directly from invariant networks, known to be universal, via our decomposition theorem. The resulting architecture differs from others in the literature, as explained in more details in Remarks 1 and 2.

## 2 Preliminaries

In this section, we review some notions of group actions and introduce invariance/equivariance for maps. In Appendix A, we summarize some necessary notions and various examples for groups.

For sets $X$ and $V$, we consider the set $\mathrm{Map}(X, V) = V^X = \{f\colon X \to V\}$ of maps from $X$ to $V$. In many situations, we regard $X$ as an index set and $V$ as a set of objects (such as channels or pictures). We show some examples.

**Example 1.** *(i) If $X = \{1, 2, \ldots, n\}$ and $V = \mathbb{R}$, then $\mathrm{Map}(X, V)$ is idenfied with $\mathbb{R}^n$ by $\mathrm{Map}(X, V) \to \mathbb{R}^n\colon f \mapsto (f(1), f(2), \ldots, f(n))^\top$.*

(ii) *Let $\ell, m$ be positive integers, and $X = \{1, 2, \ldots, \ell\} \times \{1, 2, \ldots, m\}$ and $V = \{0, 1, \ldots, 255\}^3$. Then, $\mathrm{Map}(X, V)$ can be regarded as the set of digital images of $\ell \times m$ pixels with the RGB color channels. For $f \in \mathrm{Map}(X, V)$ and $(i, j) \in X$, $f(i, j) = (r_{ij}, g_{ij}, b_{ij})^\top$ $(r_{ij}, g_{ij}, b_{ij} \in \{1, 2, \ldots, 255\})$ represents RGB color at $(i, j)$-th pixel.*

(iii) *Let $V' \subset V^X$ be a set of some digital images as in Example 1(ii) and $X' = \{1, 2, \ldots, n\}$. Then, $V'^{X'}$ is the set of $n$-tuples of the digital images in $V'$.*

(iv) *For $X = \mathbb{R}^2$ and $V = \mathbb{R}^3$, $\mathrm{Map}(\mathbb{R}^2, \mathbb{R}^3) = (\mathbb{R}^3)^{\mathbb{R}^2}$ can be regarded as the space of "ideal" images. Here, the "ideal" means that the size of the image is "infinitely extended" and the pixels of the image are "infinitely detailed" in the sense of Yarotsky (2022). This is a similar notion to the set $L^2(\mathbb{R}^\nu, \mathbb{R}^m)$ of the "signals" introduced in Yarotsky (2022, Section 3.2).*

Next, we consider a group action on the set $V^X$. Let $G$ be a group and $X$ be a $G$-set, i.e., a set on which $G$ acts from the left. Then, $G$ also acts on $V^X$ from the left[1] for any set $V$ by

$$(\sigma \cdot f)(x) = f(\sigma^{-1} \cdot x), \tag{1}$$

for $f \in V^X$ and $\sigma \in G$. For $f \in V^X$, let $O_f$ be the $G$-orbit $G \cdot f = \{\sigma \cdot f \mid \sigma \in G\}$. A few examples of such type of action are listed below.

**Example 2.** (i) *The permutation group $S_n$ of the set $X = \{1, 2, \ldots, n\}$ acts on $X$ by permutation. Any unordered set $\{v_1, v_2, \ldots, v_n\}$ of $n$ elements of $V$ can be regarded as an $S_n$-orbit of the ordered $n$-tuple $(v_1, v_2, \ldots, v_n) \in V^X$ for $X = \{1, 2, \ldots, n\}$.*

(ii) *The $S_n$-orbit of an element $f$ in $\mathrm{Map}(\{1, 2, \ldots, n\}, \mathbb{R}^3)$ can be regarded as a point cloud consisting of $n$ points in $\mathbb{R}^3$.*

(iii) *Let $X = \{1, 2, \ldots, n\}^2$ and $V = \mathbb{R}$. We consider the diagonal $S_n$-action $\sigma \cdot (i, j) = (\sigma(i), \sigma(j))$ $(\sigma \in S_n)$ on $X$. Then, $S_n$ also acts on $V^X = \mathbb{R}^{n \times n}$. Let $\mathrm{Sym}(n)$ be the subset of symmetric matrices in $\mathbb{R}^{n \times n}$. This $\mathrm{Sym}(n)$ is stable by the diagonal action of $S_n$. The $S_n$-orbit $S_n \cdot A$ of an $A \in \mathrm{Sym}(n) \subset \mathbb{R}^{n \times n}$ can be regarded as an isomorphism class of undirected weighted graph. Indeed, $\mathrm{Sym}(n)$ is the set of adjacency matrices of undirected weighted graphs, and two graphs are isomorphic if and only if the corresponding adjacency matrices $A_1, A_2$ satisfy $A_1 = \sigma \cdot A_2$ for an element $\sigma \in S_n$.*

We define invariant and equivariant maps as follows:

**Definition 1.** *Let $G$ be a group, $X, Y$ be two $G$-sets, and $V, W$ be two sets. Then, a map $F$ from $V^X$ to $W$ is $G$-invariant if $F$ satisfies $F(\sigma \cdot f) = F(f)$ for every $\sigma \in G$ and $f \in V^X$. A map $F$ from $V^X$ to $W^Y$ is $G$-equivariant if $F$ satisfies $F(\sigma \cdot f) = \sigma \cdot F(f)$ for every $\sigma \in G$ and $f \in V^X$. We denote*

$$\mathrm{Inv}_G(V^X, W) = \{F\colon V^X \to W \mid G\text{-invariant}\},$$

*and*

$$\mathrm{Equiv}_G(V^X, W^Y) = \{F\colon V^X \to W^Y \mid G\text{-equivariant}\}.$$

---

[1]Acting from the left means that for any $\sigma, \tau \in G$, the formula $\sigma \cdot (\tau \cdot f) = (\sigma\tau) \cdot f$ holds. To ensure this, we must use $\sigma^{-1}$, rather than $\sigma$, on the right-hand part of equation (1). Otherwise we would have $\sigma \cdot (\tau \cdot f) = (\tau\sigma) \cdot f$.

Some examples of invariant or equivariant tasks are the following

**Example 3.**      *(i) Let $X = Y = \{1, 2, \ldots, n\}$ and $V = \mathbb{R}^3$. The classification task of point clouds can be regarded as a task to find an appropriate $S_n$-invariant map from $V^X$ to a set of classes $W = \{c_1, c_2, \ldots, c_m\}$.*

*(ii) A task of anomaly detection from n pictures is a permutation equivariant task as in Zaheer et al. (2017, Appendix I). This is to find an appropriate $S_n$-equivariant map $V^X$ to $W^Y$ for $V = (\{0, 1, \ldots, 255\}^3)^{\ell \times m}$, $W = \{0, 1\}$, and $X = Y = \{1, 2, \ldots, n\}$.*

*(iii) A task of classification of digital images of $\ell \times \ell$ pixels is finding an appropriate 90-degree rotation invariant map $V^X$ to a set of classes $W = \{c_1, c_2, \ldots, c_m\}$, where $V = \{0, 1, \ldots, 255\}^3$ and $X = \{1, 2, \ldots, \ell\}^2$. An image segmentation task can be regarded as finding an appropriate 90-degree rotation equivariant map $V^X$ to $W^Y$ for the same $V, W, X$ as Example 3 (ii) and $Y = \{1, 2, \ldots, \ell\}^2$.*

*(iv) An extension of Example 3 (iii) to "ideal images" is finding an $\mathrm{SO}_2(\mathbb{R})$-invariant map from $V^X$ to $W$ (or $\mathrm{SO}_2(\mathbb{R})$-equivariant map from $V^X$ to $W^Y$) for and $X = \mathbb{R}^2$, $V = \mathbb{R}^3$, and $W = \mathbb{R}$ (or $X = Y = \mathbb{R}^2$, $V = \mathbb{R}^3$, and $W = \mathbb{R}$ respectively).*

## 3    Invariant-equivariant relation and universal approximation

### 3.1    Warm up example with $S_n$

We begin this section with a warm-up example. In this subsection, $G = S_n$, $X = \{1, \ldots, n\}$ and $V, W$ are generic sets. We denote as $f$ an element of $V^X$. The action of $S_n$ on $V^X$ is as in equation (1)

$$(\sigma \cdot f)(i) = f(\sigma^{-1}(i)), \ \forall \ 1 \le i \le n, \tag{2}$$

and so is the action of $S_n$ on $W^X$. Let $F$ be a map from $V^X$ to $W^X$, then there exist $n$ maps $F_1, \ldots, F_n$ from $V^X$ to $W$ such that

$$F(f)(i) = F_i(f), \ \forall \ 1 \le i \le n. \tag{3}$$

$F_i$ is just a shorter notation for the map $F(\cdot)(i)$. Now let us make the assumption that $F$ belongs to $\mathrm{Equiv}_{S_n}(V^X, W^X)$, the equivariance property $F(\sigma \cdot f) = \sigma \cdot (F(f))$ implies for all $i$,

$$F_i(\sigma \cdot f) = F_{\sigma^{-1}(i)}(f). \tag{4}$$

Let us focus on $i = 1$. Notice that if $\sigma$ is a permutation such $\sigma(1) = 1$, it is straightforward from equation (4) that $F_1(\sigma \cdot f) = F_1(f)$ for all $f \in V^X$. Since it is fairly known that the set of permutations such that $\sigma(1) = 1$ is a subgroup of $S_n$, denoted as $\mathrm{Stab}_{S_n}(1)$, (this group is isomorphic to $S_{n-1}$), we deduce from equation (4) that

$$F_1 = F(\cdot)(1) \in \mathrm{Inv}_{\mathrm{Stab}_{S_n}(1)}(V^X, W). \tag{5}$$

Moreover, if we let $(1 \ i)$ be the transposition that exchanges 1 and $i$ and fix any other $j \in X$, we obtain easily from equation (4) that for all $f$

$$F_1((1 \ i) \cdot f) = F_{(1 \ i) \cdot 1}(f) = F_i(f). \tag{6}$$

To sum up, we have almost proven the following claim.

**Proposition 1.** *$F \in \mathrm{Equiv}_{S_n}(V^X, W^X)$ if and only if there exists $F_1 \in \mathrm{Inv}_{\mathrm{Stab}_{S_n}(1)}(V^X, W)$ such that for all $f \in V^X$, for $i = 1, \ldots, n$,*

$$F(f)(i) = F_1((1 \ i) \cdot f). \tag{7}$$

*Proof.* The discussion above is a proof of the nontrivial implication. Reciprocally, one easily verifies that picking $F_1 \in \mathrm{Inv}_{\mathrm{Stab}_{S_n}(1)}(V^X, W)$ and defining $F : V^X \to W^X$ by

$$F(f)(i) = F_1((1 \ i) \cdot f,$$

yields a map $F \in \mathrm{Equiv}_{S_n}(V^X, W^X)$.     $\square$

The rest of this section is dedicated to a generalization of this result to a generic group $G$.

### 3.2 Invariant-equivariant relation

Our main theorem is a relation between a $G$-equivariant maps and some invariant maps for some subgroups of $G$. Recall that, for any $y$ in a $G$-set $Y$, its stabilizer, denoted as $\mathrm{Stab}_G(y)$, is the subgroup of all the $\sigma \in G$ such that $\sigma \cdot y = y$ (see the recall en groups and group actions in Appendix A).

**Theorem 1.** *Let $G$ be a group and $X, Y$ be two $G$-sets. Let $V, W$ be two sets. Let $Y = \bigsqcup_{i \in I} O_{y_i}$ be the $G$-orbit decomposition of $Y$. We fix a system of representatives $\{y_i \in Y \mid i \in I\}$. Then, the following map is bijective:*

$$\Phi : \left| \begin{array}{ccc} \mathrm{Equiv}_G(V^X, W^Y) & \longrightarrow & \prod_{i \in I} \mathrm{Inv}_{\mathrm{Stab}_G(y_i)}(V^X, W) \\[2mm] F & \longmapsto & \Phi(F) = (F(\cdot)(y_i))_{i \in I}. \end{array} \right.$$

*Moreover, its inverse map*

$$\Psi : \left| \begin{array}{ccc} \prod_{i \in I} \mathrm{Inv}_{\mathrm{Stab}_G(y_i)}(V^X, W) & \longrightarrow & \mathrm{Equiv}_G(V^X, W^Y) \\[2mm] (\widetilde{F}_i)_{i \in I} & \longmapsto & \Psi\left( (\widetilde{F}_i)_{i \in I} \right), \end{array} \right.$$

*is defined by*

$$\Psi\left( (\widetilde{F}_i)_{i \in I} \right)(f)(y) = \widetilde{F}_i(\sigma \cdot f), \tag{8}$$

*for any $f \in V^X$ and $y \in Y$ such that $y \in O_{y_i}$ and $y = \sigma^{-1} \cdot y_i$ for some $\sigma \in G$.*

*In addition, if $V$ and $W$ are vector spaces over $\mathbb{R}$, then $V^X$, $W^Y$, $\mathrm{Equiv}_G(V^X, W^Y)$, and $\prod_{i \in I} \mathrm{Inv}_{\mathrm{Stab}_G(y_i)}(V^X, W)$ are also vector spaces over $\mathbb{R}$ and $\Phi$ and $\Psi$ are $\mathbb{R}$-linear isomorphisms.*

*Proof.* To prove Theorem 1, we shall prove that the maps $\Phi$ and $\Psi$ are well-defined and these maps are the inverse maps of each other, i.e., $\Psi \circ \Phi$ is the identity on $\mathrm{Equiv}_G(V^X, W^Y)$ and $\Phi \circ \Psi$ is the identity on $\prod_{i \in I} \mathrm{Inv}_{\mathrm{Stab}_G(y_i)}(V^X, W)$.

We first show the well-definedness of $\Phi$. That is, for $\Phi(F) = (F(\cdot)(y_i))_{i \in I}$, we shall prove that $F(\cdot)(y_i) : V^X \to W$ is $\mathrm{Stab}_G(y_i)$-invariant for any $i \in I$. To do so, we check that for $\sigma \in \mathrm{Stab}_G(y_i)$ and $f \in V^X$, $F(\sigma \cdot f)(y_i) = F(f)(y_i)$. Let $F$ be a $G$-equivariant map from $V^X$ to $W^Y$ and $\sigma$ be an element in $\mathrm{Stab}_G(y_i)$. This implies $\sigma^{-1} \cdot y_i = y_i$, hence the inverse $\sigma^{-1}$ is also in $\mathrm{Stab}_G(y_i)$. Then, by $G$-equivariance of $F$, we have

$$F(\sigma \cdot f)(y_i) = (\sigma \cdot F)(f)(y_i) = F(f)(\sigma^{-1} \cdot y_i) = F(f)(y_i).$$

Hence, the map $F(\cdot)(y_i)$ is $\mathrm{Stab}_G(y_i)$-invariant. This implies that the map

$$\Phi : \mathrm{Equiv}_G(V^X, W^Y) \to \prod_{i \in I} \mathrm{Inv}_{\mathrm{Stab}_G(y_i)}(V^X, W); F \mapsto (F(\cdot)(y_i))_{i \in I}$$

is well-defined.

Next, we prove that the map $\Psi$ is well-defined. The well-definedness of $\Psi$ can be rephrased that the image $\Psi((\widetilde{F}_i)_{i \in I})(f)(y) = \widetilde{F}_i(\tau \cdot f)$ for $\tau \in G$ such that $y = \tau^{-1} \cdot y_i$ of $(\widetilde{F}_i)_{i \in I} \in \prod_{i \in I} \mathrm{Inv}_{\mathrm{Stab}_G(y_i)}(V^X, W)$ is independent of the choice of $\tau \in G$ and is $G$-equivariant. We first notice that such a $\tau$ exists since, from the $G$-orbit decomposition $Y = \bigsqcup_{i \in I} O_{y_i}$, there is an orbit representative $y_i$ such that $y \in O_{y_i}$, and thus $y = \tau^{-1} \cdot y_i$ for some $\tau \in G$. If $\tau' \in G$ is another choice so that $y = \tau'^{-1} \cdot y_i$, then we have $\tau'^{-1} \cdot y_i = y = \tau^{-1} \cdot y_i$. This implies that $\tau \tau'^{-1}$ stabilizes $y_i$. Thus, we can represent $\tau' = \sigma \tau$ for some $\sigma \in \mathrm{Stab}_G(y_i)$. The value of $\widetilde{F}_i$ at $\tau' \cdot f$ becomes

$$\widetilde{F}_i(\tau' \cdot f) = \widetilde{F}_i((\sigma \tau) \cdot f)) = \widetilde{F}_i(\sigma \cdot (\tau \cdot f)) = \widetilde{F}_i(\tau \cdot f).$$

Here, the last equality is deduced from $\mathrm{Stab}_G(y_i)$-invariance of $\widetilde{F}_i$. This implies that the definition of the value $\Psi(\widetilde{F}_i)(f)(y)$ is independent of the choice of $\tau \in G$ satisfying $y_i = \tau \cdot y$.

We set $F = \Psi((\widetilde{F}_i)_{i\in I})$. To show $G$-equivariance of $F$, it is sufficient to check that for any $\sigma \in G$, any $f \in V^X$, and any $y \in Y$, $\sigma \cdot F(f)(y) = F(\sigma \cdot f)(y)$ holds. Let $y \in O_{y_i}$ and $y = \tau^{-1} \cdot y_i$ for some $\tau \in G$. Then, because $\sigma^{-1} \cdot y = (\sigma^{-1}\tau^{-1}) \cdot y_i \in O_{y_i}$,

$$\text{(LHS)} = \sigma \cdot F(f)(y) = F(f)(\sigma^{-1} \cdot y) = F(f)((\sigma^{-1}\tau^{-1}) \cdot y_i)$$
$$= F(f)((\tau\sigma)^{-1} \cdot y_i) = \widetilde{F}_i((\tau\sigma) \cdot f).$$

Here, the last equality follows from the definition of the map of $\Psi$ in equation (8). On the other hand, we have

$$\text{(RHS)} = F(\sigma \cdot f)(y) = F(\sigma \cdot f)(\tau^{-1} \cdot y_i) = \widetilde{F}_i(\tau \cdot (\sigma \cdot f)) = \widetilde{F}_i((\tau\sigma) \cdot f).$$

Here, the third equality also follows from the definition of the map of $\Psi$ in equation (8). Therefore, $F$ satisfies $\sigma \cdot F(f)(y) = F(\sigma \cdot f)(y)$ for any $\sigma \in G$, any $f \in V^X$, and any $y \in Y$. Hence, $F$ is $G$-equivariant.

Finally, we show that the map $\Psi$ is the inverse map of the map $\Phi$ and vice versa, i.e., both $\Psi \circ \Phi$ and $\Phi \circ \Psi$ are identities. Let $F$ be a map in $\mathrm{Equiv}_G(V^X, W^Y)$. Then, the image of the map $\Phi$ can be written as $(\widetilde{F}_i)_{i\in I}$ such that $\widetilde{F}_i(f) = F(f)(y_i)$. Then, the map $\Psi$ takes $(\widetilde{F}_i)_{i\in I}$ to $F'$ such that $F'(f)(y) = \widetilde{F}_i(\tau \cdot f) = F(\tau \cdot f)(y_i)$ for $y \in Y$ and $\tau \in G$ such that $y = \tau^{-1} \cdot y_i$. Because $F$ is $G$-equivariant, we have

$$F(f)(y) = F(f)(\tau^{-1} \cdot y_i) = (\tau \cdot F)(f)(y_i)$$
$$= F(\tau \cdot f)(y_i) = F'(f)(y) = \Psi \circ \Phi(F(f))(y).$$

Hence, $F = F' = (\Psi \circ \Phi)(F)$ holds for any $F \in \mathrm{Equiv}_G(V^X, W^Y)$. In particular, $\Psi \circ \Phi$ is the identity.

Conversely, $(\widetilde{F}_i)_{i\in I} \in \prod_{i\in I} \mathrm{Inv}_{\mathrm{Stab}_G(y_i)}(V^X, W)$ is given. Let $F = \Psi((\widetilde{F}_i)_{i\in I})$. Then, the image of $F$ by $\Phi$ becomes $\Phi(F)(f) = (F(f)(y_i))_{i\in I}$. Now, for $i_0 \in I$,

$$F(f)(y_{i_0}) = \Psi((\widetilde{F}_i)_{i\in I})(f)(y_{i_0}) = \widetilde{F}_{i_0}(\tau \cdot f)$$

for a $\tau \in G$ satisfying $y_{i_0} = \tau \cdot y_{i_0}$. In particular, $\tau$ is in $\mathrm{Stab}_G(y_{i_0})$. By $\mathrm{Stab}_G(y_{i_0})$-invariance of $\widetilde{F}_{i_0}$, $\widetilde{F}_{i_0}(\tau \cdot f) = \widetilde{F}_{i_0}(f)$ holds. Thus, we have

$$(\Phi \circ \Psi)((\widetilde{F}_j)_{j\in I})(f) = \Phi(\Psi((\widetilde{F}_j)_{j\in I})(f)) = \{\Psi((\widetilde{F}_j)_{j\in I})(f)(y_i)\}_{i\in I}$$
$$= (\widetilde{F}_i(f))_{i\in I} = (\widetilde{F}_i)_{i\in I}(f).$$

Therefore, the composition $\Phi \circ \Psi$ is the identity.

Finally, we justify the final statement of the theorem in the case $V$ and $W$ are real vector spaces. Under this assumption, it is fairly known that $V^X$, $W^Y$, $\mathrm{Map}(V^X, W^Y)$ as well as $\mathrm{Map}(V^X, W)$ are real vector spaces. Therefore, $\prod_{i\in I} \mathrm{Map}(V^X, W)$ is also a vector space as the Cartesian product of vector spaces. By an abuse of notation, consider $\Phi$ as a map between $\mathrm{Map}(V^X, W^Y)$ and $\prod_{i\in I} \mathrm{Map}(V^X, W)$, and let us show that it is linear.

Let $F, F' \in \mathrm{Map}(V^X, W^X)$ and $\lambda \in \mathbb{R}$. It is clear that for all $f \in V^X$, $(F + \lambda F')(f) = F(f) + \lambda F'(f)$ and thus for all $y \in Y$, $(F + \lambda F')(f)(y) = F(f)(y) + \lambda F'(f)(y)$. Hence,

$$\Phi(F + \lambda F') = ((F + \lambda F')(\cdot)(y_i))_{i\in I}$$
$$= (F(\cdot)(y_i) + \lambda F'(\cdot)(y_i))_{i\in I}$$
$$= (F(\cdot)(y_i))_{i\in I} + \lambda (F'(\cdot)(y_i))_{i\in I},$$

Which justifies the linearity of $\Phi : \mathrm{Map}(V^X, W^Y) \to \prod_{i\in I} \mathrm{Map}(V^X, W)$.

Next, we shall show that $\mathrm{Equiv}_G(V^X, W^Y)$ is a linear subspace of $\mathrm{Map}(V^X, W^Y)$. In order to do that, we first have to show that the action of $F'$ on $\mathrm{Map}(V^X, W^Y)$ is itself linear. Let $F, F' \in \mathrm{Map}(V^X, W^Y)$ and

$\lambda \in \mathbb{R}$, for any $f \in V^X$, any $y \in Y$, and any $\sigma \in G$, we have :

$$\begin{aligned} \sigma \cdot (F + \lambda F')(f)(y) &= (F(f) + \lambda F'(f))(\sigma^{-1} \cdot y) \\ &= F(f)(\sigma^{-1} \cdot y) + \lambda F'(f)(\sigma^{-1} \cdot y) \\ &= \sigma \cdot F(f)(y) + \lambda \sigma \cdot F'(f)(y), \end{aligned}$$

Which yields $\sigma \cdot (F + \lambda F') = \sigma \cdot F + \lambda \sigma \cdot F'$. That being done, we go back on showing that $\mathrm{Equiv}_G(V^X, W^Y)$ is a linear subspace of $\mathrm{Map}(V^X, W^Y)$. Let $F, F' \in \mathrm{Equiv}_G(V^X, W^Y)$ and $\lambda \in \mathbb{R}$, for any $\sigma \in G$, we have

$$\begin{aligned} \sigma \cdot (F + \lambda F')(f) &= (\sigma \cdot F + \lambda \sigma \cdot F')(f) \\ &= \sigma \cdot F(f) + \lambda \sigma \cdot F'(f) \\ &= F(\sigma \cdot f) + \lambda F'(\sigma \cdot f) \\ &= (F + \lambda F')(\sigma \cdot f), \end{aligned}$$

where the third equality is due to the $G$-equivariance property. Hence, since the null map is clearly equivariant too, this makes $\mathrm{Equiv}_G(V^X, W^Y)$ a real vector subspace of $\mathrm{Map}(V^X, W^Y)$.

Consequently, the image of $\mathrm{Equiv}_G(V^X, W^Y)$ by $\Psi$ must be a vector subspace of $\prod_{i \in I} \mathrm{Map}(V^X, W)$. Since we have proven earlier that $\Phi : \mathrm{Equiv}_G(V^X, W^Y) \to \prod_{i \in I} \mathrm{Inv}_{\mathrm{Stab}_G(y_i)}(V^X, W)$ is a bijection, in particular, it is a surjection. Hence, the aforementioned image is $\prod_{i \in I} \mathrm{Inv}_{\mathrm{Stab}_G(y_i)}(V^X, W)$, therefore it is a vector space.

To conclude, notice that we have shown that $\Phi : \mathrm{Equiv}_G(V^X, W^Y) \to \prod_{i \in I} \mathrm{Inv}_{\mathrm{Stab}_G(y_i)}(V^X, W)$ is a linear bijection between vector spaces. Thus, by a fairly known fact from linear algebra, its inverse $\Psi$ must be linear too. $\qquad \square$

Theorem 1 implies that any $G$-equivariant map $F$ is determined by the $\mathrm{Stab}_G(y_i)$-invariant maps $F(\cdot)(y_i)$ for $i \in I$. In other words, the parts $F(\cdot)(y)$ for $y \notin \{y_i \mid i \in I\}$ are redundant to construct $G$-equivariant map $F$.

This theorem is quite elementary, in the sense that its statement as well as its proof, only requires basic knowledge in group theory. However, the idea of linking $G$-equivariant maps to invariant maps on some subgroups of $G$ is not new and has more profound implications. For instance, it is central in group representation theory as it relates to the so-called Frobenius reciprocity theorem (Curtis & Reiner, 1966), about the relationships between the representation of $G$ and the representations of its subgroups. In addition, the result from Theorem 1 can be recovered from some other more general theorems in more advanced topics, such as Theorem 1.1.4 in Cap & Slovák (2009), about homogeneous vector bundles.

### 3.3 Construction of universal approximators

As an application of Theorem 1, we construct universal approximators for continuous $G$-equivariant maps. In this section, we assume that $V$ and $W$ are normed vector spaces over $\mathbb{R}$ whose norms are $\|\cdot\|_V$ and $\|\cdot\|_W$ respectively. Then, we define norms on $V^X$ and $W^Y$ by the supremum norms

$$\|f\|_{V^X} = \sup_{x \in X} \|f(x)\|_V. \tag{9}$$

for $f \in V^X$ and similarly for $g \in W^Y$. Let $K \subset V^X$ be a compact subset. We assume that $K$ is stable by the action of $G$, i.e., $\sigma \cdot f \in K$ for any $\sigma \in G$ and $f \in K$. We denote $\mathrm{Equiv}_G^{\mathrm{cont}}(K, W^Y)$ (resp. $\mathrm{Inv}_H^{\mathrm{cont}}(K, W)$ for a subgroup $H$) the subset of continuous maps in $\mathrm{Equiv}_G(K, W^Y)$ (resp. $\mathrm{Inv}_H(K, W)$):

$$\mathrm{Equiv}_G^{\mathrm{cont}}(K, W^Y) = \{F \in \mathrm{Equiv}_G(K, W^Y) \mid \text{continuous}\}$$

$$\mathrm{Inv}_H^{\mathrm{cont}}(K, W) = \{\widetilde{F} \in \mathrm{Inv}_H(K, W) \mid \text{continuous}\}$$

For a subgroup $H \subset G$, let $\mathcal{H}_{H\text{-inv}}$ be a hypothesis set in $\mathrm{Inv}_H^{\mathrm{cont}}(K, W)$ consisting of some deep neural networks. We assume that any maps in $\mathrm{Inv}_H^{\mathrm{cont}}(K, W)$ can be approximated by an element in $\mathcal{H}_{H\text{-inv}}$, *i.e.*,

for any $\widetilde{F} \in \mathrm{Inv}_H^{\mathrm{cont}}(K, W)$ and any $\varepsilon > 0$, there is an element $\widehat{\widetilde{F}} \in \mathcal{H}_{H\text{-inv}}$ such that

$$\sup_{f \in K} \|\widehat{\widetilde{F}}(f) - \widetilde{F}(f)\|_W < \varepsilon.$$

We define a hypothesis set $\mathcal{H}_{G\text{-equiv}}$ in $\mathrm{Equiv}_G(K, W^Y)$ by aggregating the $W$-valued hypothesis sets $\mathcal{H}_{\mathrm{Stab}_G(y_i)\text{-inv}}$ as follows:

$$\mathcal{H}_{G\text{-equiv}} = \Psi\left(\prod_{i \in I} \mathcal{H}_{\mathrm{Stab}_G(y_i)\text{-inv}}\right), \tag{10}$$

where $\Psi$ is the map defined in Theorem 1. Then, the set $\mathcal{H}_{G\text{-equiv}}$ is included in $\mathrm{Equiv}_G^{\mathrm{cont}}(K, W^Y)$. Indeed, by Theorem 1, this set is included in $\mathrm{Equiv}_G(K, W^Y)$ and we can show that $F = (F_y)_{y \in Y} \colon K \to W^Y$ is continuous if and only if $F_y \colon K \to W$ is continuous for all $y \in Y$ because we consider the supremum norm on $W^Y$.

The following theorem constructs a universal approximator for $G$-equivariant map from some invariant universal approximators:

**Theorem 2.** *Any element in* $\mathrm{Equiv}_G^{\mathrm{cont}}(K, W^Y)$ *can be approximated by an element of* $\mathcal{H}_{G\text{-equiv}}$ *defined in equation (10). More precisely, for any* $F \in \mathrm{Equiv}_G^{\mathrm{cont}}(K, W^Y)$ *and any* $\varepsilon > 0$, *there exists an element in* $\widehat{F} \in \mathcal{H}_{G\text{-equiv}}$ *such that*

$$\sup_{f \in K} \|\widehat{F}(f) - F(f)\|_{W^Y} \leq \varepsilon.$$

*Proof.* By the definition of the hypothesis set $\mathcal{H}_{\mathrm{Stab}_G(y_i)\text{-inv}}$, for any $\varepsilon > 0$ and any $i \in I$, there is an element $\widehat{\widetilde{F}}_i \in \mathcal{H}_{\mathrm{Stab}_G(y_i)\text{-inv}}$ such that

$$\sup_{f \in K} \|\widehat{\widetilde{F}}_i(f) - \widetilde{F}_i(f)\|_W \leq \varepsilon \tag{11}$$

We set $\widehat{F} = \Psi((\widehat{\widetilde{F}}_i)_{i \in I})$. The map $F$ can also be written as $F = \Psi((\widetilde{F}_i)_{i \in I})$. Then, the norm of the difference between $\widehat{F}$ and $F$ for $f \in K$ at $y \in Y$ becomes as follows: If $y \in O_{y_i}$ and $y = \sigma^{-1} \cdot y_i$ for $\sigma \in G$, then

$$\begin{aligned}
\|\widehat{F}(f)(y) - F(f)(y)\|_W &= \|\widehat{F}(f)(\sigma^{-1} \cdot y_i) - F(f)(\sigma^{-1} \cdot y_i)\|_W \\
&= \|\widehat{F}(\sigma \cdot f)(y_i) - F(\sigma \cdot f)(y_i)\|_W \\
&= \|\widehat{\widetilde{F}}_i(\sigma \cdot f) - \widetilde{F}_i(\sigma \cdot f)\|_W \\
&\leq \sup_{f \in K} \|\widehat{\widetilde{F}}_i(\sigma \cdot f) - \widetilde{F}_i(\sigma \cdot f)\|_W \\
&\leq \sup_{f \in K} \|\widehat{\widetilde{F}}_i(f) - \widetilde{F}_i(f)\|_W \leq \varepsilon
\end{aligned}$$

The second inequality follows from the stability of $K$ by the action of $G$, and the last inequality follows from inequality equation (11).

Hence, the difference between $\widehat{F}$ and $F$ at $f \in K$ can be written as

$$\|\widehat{F}(f) - F(f)\|_{W^Y} = \sup_{y \in Y} \|\widehat{F}(f)(y) - F(f)(y)\|_W \leq \varepsilon. \tag{12}$$

This completes the proof because $f \in K$ is arbitrary in the above argument. $\qquad\square$

The conclusion of this Theorem relies on the assumption that some universal classes $\mathcal{H}_{H\text{-inv}}$ of invariant maps do exist for the subgroups of $G$. We argue that this assumption is always fulfilled. Indeed, given any

universal class with *a priori* no symmetry, for instance, usual multi-layers perceptrons, one can always turn it into a universal class of invariant maps by simple group averaging, as remarked by Yarotsky (2022).

However, such straightforward constructions are clearly unrealistic for large groups like $S_n$. The interest of Theorem 2 is to enable reducing the study of efficient $G$-equivariant universal architecture to the study of $\mathrm{Stab}_G(y_i)$-invariant ones. It offers an alternative to the equivariant or invariant "symmetrization-based" constructions in Yarotsky (2022, Section 2.1) by group averaging.

### 3.3.1 On the structural characteristics of universal equivariant networks arising from Theorem 2

**On the $G$-action between the hidden layers.** Several works have studied the design of multi-layer universal equivariant architectures, including Zaheer et al. (2017), Maron et al. (2019b), Segol & Lipman (2019), and Ravanbakhsh (2020). How do a universal architecture arising from our Theorem 2, differ from the already existing aforementioned ones?

The main difference resides in the nature of the $G$-action in the hidden layers. In fact the way our architecture is built involves a new $G$-action, noted "$*$", which is different from the original action "$\cdot$". We provide below an explanation of this phenomenon in the form of a remark. The rigorous demonstrations are left to Appendix B and require some background in group Representation Theory.

**Remark 1.** *Say that we seek to design multi-layer architectures that are equivariant to some $G$-action "$\cdot$". The usual strategy, as in the case of the previously mentioned works, is to focus on designing a single layer block that is equivariant for "$\cdot$". Then by stacking these blocks, we obtain an equivariant multi-layer architecture, because equivariance is preserved by composition.*

*Now, consider the scenario of our construction from Theorem 2. Assume that we are given some families of multi-layers $H_i$-invariant maps, for the action "$\cdot$", where the $H_i \subset G$ are some subgroups that have been identified thanks to the decomposition from Theorem 1. Then, by "aggregating" some of these $H_i$-invariant maps, and applying $\Psi$, as is equation (10), a new multi-layer architecture $\widehat{F}$ is built. Moreover, by Theorem 1 again, the $\widehat{F}$ obtained by this procedure are guaranteed to be equivariant for the action "$\cdot$".*

*What can be said about the layer-wise structure of such $\widehat{F}$? It turns out that the $G$-equivariance of the hidden layers is no longer realized by "$\cdot$" on both the input and the first hidden space. Instead, it involves another $G$-action "$*$" in the hidden space. If we look at the first layer $\varphi_1$, which consists in "plugging-in" the "aggregated" invariant maps to the input space (see Figure 2), its equivariance, from the input layer to the first hidden layer of $\widehat{F}$, will be written as $\varphi_1(\sigma \cdot f) = \sigma * \varphi_1(f)$, as seen in Figure 1(b).*

*Overall, using commutative diagrams, Figure 1 highlights the differences between the group actions involved in these architectures.*

*This new action "$*$", which is still an action of the group $G$, can be described as the "induced representation of the restricted representation action on the hidden layers". In Appendix B, a rigorous discussion about this phenomenon and definition of this action "$*$" are done using tools from Representation Theory.*

**On the number of free parameters.** Another notorious advantage of multi-layer equivariant architectures such as Zaheer et al. (2017); Maron et al. (2020), is that the design of equivariant blocks requires few free parameters thanks to a phenomenon of parameters sharing enabled by the symmetry. Hence, in terms of the total number of free parameters, there is a significant advantage in using layer-wise equivariant networks, rather than naive fully connected ones, in order to approximate an equivariant map. Is it also the case for our architecture?

We clarify that point in the following remark. We claim that, indeed, our equivariant architecture is more efficient than naive fully connected networks, in terms of a number of free parameters. Proofs and rigorous explanations are left in appendix C.

**Remark 2.** *Although our equivariant model differs from the standard equivariant models, the number of parameters of our models can also be less than that of a fully connected neural network. For example, let $G = S_n$ act on $X = Y = \{1, \ldots, n\}$ by permutation and $V, W$ be vector spaces over $\mathbb{R}$. For $S_n$-equivariant map $F \colon V^n \to W^n$, we can take $G$-equivariant deep neural network $\widehat{F}$ approximating $F$ by Theorem 2. Then,*

(a) Usual equivariant architecture: the action "$\cdot$" is the same in the input space as well as in each hidden spaces.

(b) Equivariant architecture arising from Theorem 2: the action "$\cdot$" on the input space is different than the action "$*$", in red, on the hidden spaces.

Figure 1: Commutative diagrams for comparing between a usual equivariant multi-layer architecture, and ours arising from Theorem 2. For each diagram, the two lines both represent the same multi-layer equivariant architecture with the $\varphi_i$ being the equivariant layers between hidden spaces. Each square encapsulating a $\circlearrowright$ symbol is a commutative diagram that represents the equivariance of the layer $\varphi_i$, *i.e*, the fact that $\varphi_i$ commutes with some action of the group. This means that, within these squares, from the top left to the bottom right the two possible paths are equal.

*the first hidden layer of $\widehat{F}$ can be written as the form $V_1^{n^2}$ for a vector space $V_1$. The linear map $V^n \to V_1^{n^2}$ is equivariant for permutation action on $V^n$ and the direct sum of "induced representations of restriction representations to the stabilizer subgroup" on $V_1^{n^2}$. By the argument on irreducible representations, the number of parameters of the linear map between the input layer $V^n$ and the first hidden layer $V_1^{n^2}$ becomes $5 \dim V \dim V_1$. Because the number of parameters of the linear map $V^n \to V_1^{n^2}$ for the fully connected model is $n^3 \dim V \dim V_1$, our model can be constructed by much fewer parameters than the fully connected model. In Appendix C, precise statement and proof of this fact are concluded using Schur's Lemma and Young's diagrams.*

## 4 Some specific cases

In this section, we propose an application of Theorems 1 and 2 to a few examples.

### 4.1 Permutation equivariant maps

As a first example, we consider $S_n$-equivariant maps. Recall the context from Section 3.1 where $X = Y = \{1, 2, \ldots, n\}$ and $V, W$ are generic sets. The set $Y$ only has a single orbit by the $S_n$ action: $O_1 = S_n \cdot 1$, which can be written as

$$O_1 = \{(1\ i) \cdot 1 \mid i = 1, \ldots, n\} = \{1, 2, \ldots, n\},$$

where $(i\ j) \in S_n$ is the transposition between $i$ and $j$.

Hence, we recover Proposition 1 from Theorem 1. Namely, that an $S_n$-equivariant map $F \colon V^X \to W^X$ is uniquely determined by its Stab(1)-invariant component $\widetilde{F}_1 \colon V^X \to W$ via

$$F(f) = (\widetilde{F}_1((1\ i) \cdot f))_{i=1}^n.$$

By this argument, any analysis of $S_n$-equivariance of the map $F$ can be reduced to analyze only one $\mathrm{Stab}_{S_n}(1)$-invariant map $\widetilde{F}_1 \colon V^X \to W$ (Figure 2).

Figure 2: The case of $G = S_n$ and $X = Y = \{1, 2, \ldots, n\}$. The $S_n$-orbit of $1 \in Y$ is the whole set $Y$, thus $S_n$-equivariant map $F$ is determined by only one $\mathrm{Stab}_{S_n}(1)$-invariant map $\widetilde{F}_1$.

In this case, as a hypothesis set $\mathcal{H}_{\mathrm{Stab}_{S_n}(1)}$ of $\mathrm{Stab}_{S_n}(1)$-invariant deep neural networks, we can use the universal approximators defined in Maron et al. (2019b). By this $\mathcal{H}_{\mathrm{Stab}_{S_n}(1)}$ and Theorem 2, we obtain universal approximators for $S_n$-equivariant maps.

## 4.2 Finite groups

In this subsection, $G$ is any finite group. By Proposition 3 in Appendix A, $G$ is a subgroup of $S_n$ for some $n$. Hence, $G$ acts on $X = Y = \{1, 2, \ldots, n\}$ via permutation action of $S_n$.

The $G$-orbit decomposition of $Y$ is generally nontrivial. Let $Y = \bigsqcup_{i=1}^{k} O_{y_i}$ be the $G$-orbit decomposition of $Y$ and $y_1, y_2, \ldots, y_l$ be elements of $G$ such that $O_{y_i} = G y_i$ for $i = 1, 2, \ldots, k$. We represent $O_{y_i}$ as $O_{y_i} = \{y_{i1}, y_{i2}, \ldots, y_{im_i}\}$, and choose $\sigma_{ij} \in G$ such that $y_{ij} = \sigma_{ij}^{-1} \cdot y_i$. Then, we remark that $\sum_{i=1}^{k} m_i = n$.

We consider a $G$-equivariant map $F$ from $V^X$ to $W^Y$ for vector spaces $V, W$ over $\mathbb{R}$. By Theorem 1, $F$ can be written by

$$F(f) = \left(\widetilde{F}_i(\sigma_{ij} \cdot f)\right)_{\substack{i=1,\ldots,k \\ j=1,\ldots,m_i}}, \tag{13}$$

for any $f \in V^X$, where $\widetilde{F}_i(f) = F(f)(y_i)$ is $\mathrm{Stab}_G(y_i)$-invariant. We remark that this map $F$ is determined by the $\mathrm{Stab}_G(y_i)$-invariant map $\widetilde{F}_i$ from $V^X$ to $W$ for $i = 1, 2, \ldots, l$.

For general finite group $G$, as a hypothesis space $\mathcal{H}_{\mathrm{Stab}_G(y_i)\text{-inv}}$, we choose the set of models by using deep neural networks with tensors introduced in Maron et al. (2019b), which is guaranteed to approximate any $\mathrm{Stab}_G(y_i)$-invariant continuous maps. Keriven & Peyré (2019) also construct a universal approximator for graph neural networks.

## 4.3 The special orthogonal group $\mathrm{SO}_2(\mathbb{R})$

As for Example 3 (iii), we consider the rotation actions on the set of the ideal images. The index sets are $X = Y = \mathbb{R}^2$. We consider $\mathrm{Map}(X, \mathbb{R}^3) = (\mathbb{R}^3)^X$ as the set of ideal images with the values for ideal RGB color channels $V = \mathbb{R}^3$. Let $W$ be a set of output labels. Then, the 2-dimensional special orthogonal group $\mathrm{SO}_2(\mathbb{R})$ acts on $X = Y = \mathbb{R}^2$. We consider an $\mathrm{SO}_2(\mathbb{R})$-equivariant (i.e., rotation equivariant) map $F: V^{\mathbb{R}^2} \to W^{\mathbb{R}^2}$. Then, the orbit $O_{y_i} = \mathrm{SO}_2(\mathbb{R}) \cdot y_i$ of $y_i = (i, 0) \in \mathbb{R}^2$ for $i \in I = [0, \infty)$ is $O_{y_i} = \{(a, b) \in \mathbb{R}^2 \mid a^2 + b^2 = i^2\}$,

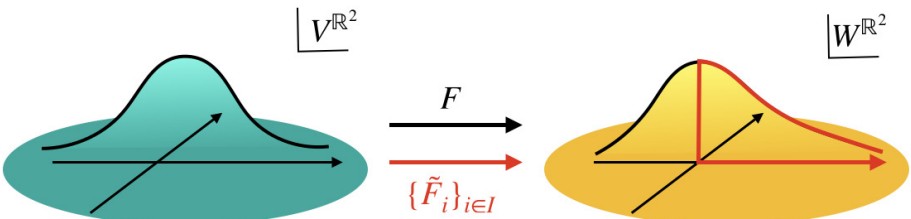

Figure 3: Rotation equivariant map $F$ is determined the maps $\{\widetilde{F}_i\}_{i \in I}$ where $I$ is the nonnegative real line $\mathbb{R}_{\geq 0}$. We can recover $F$ from $\{\widetilde{F}_i\}_{i \in I}$ by "rotating" $\{\widetilde{F}_i(g \cdot)\}_{i \in I, g \in \mathrm{SO}_2(\mathbb{R})}$
.

and $Y = \bigsqcup_{i \in I} O_{y_i}$ holds. On the other hand, the stabilizer subgroup of $\mathrm{SO}_2(\mathbb{R})$ for $y_i = (i, 0)$ is

$$\mathrm{Stab}_{\mathrm{SO}_2(\mathbb{R})}(y_i) = \begin{cases} \left\{ \begin{pmatrix} 1 & 0 \\ 0 & 1 \end{pmatrix} \right\} & \text{if } i > 0, \\ \mathrm{SO}_2(\mathbb{R}) & \text{if } i = 0. \end{cases}$$

Then, by Theorem 1, the $\mathrm{SO}_2(\mathbb{R})$-equivariant map $F$ can be written by

$$F(f) = \left( (\widetilde{F}_i(g \cdot f))_{\substack{i \in (0, \infty) \\ g \in \mathrm{SO}_2(\mathbb{R})}}, \widetilde{F}_0(f) \right).$$

This means that $\mathrm{SO}_2(\mathbb{R})$-equivariant map $F$ is determined by the maps $\widetilde{F}_i \colon ([0,1]^3)^{\mathbb{R}^2} \to W$ for $i \in [0, \infty)$ as Figure 3. Unfortunately, $\widetilde{F}_i$ for $i > 0$ does not have any invariance because the stabilizer subgroup of $\mathrm{SO}_2(\mathbb{R})$ for $(i, 0)$ is $\left\{ \begin{pmatrix} 1 & 0 \\ 0 & 1 \end{pmatrix} \right\}$.

We assume that $X, Y$ are the disk $D = \{(x, y) \mid x^2 + y^2 \leq R\}$ of radius $R > 0$. We can apply Theorem 2. Then, if there is a universal approximators $(\widehat{\widetilde{F}}_i)_{i \in I}$ of $(\widetilde{F}_i)_{i \in I}$, $\Psi((\widehat{\widetilde{F}}_i)_{i \in I})$ approximates the map $F = \Psi((\widetilde{F}_i)_{i \in I})$.

In this case, as a hypothesis set $\mathcal{H}_{\mathrm{Stab}_{\mathrm{SO}_2(\mathbb{R})}(y_i)\text{-}inv}$ of $\mathrm{Stab}_{\mathrm{SO}_2(\mathbb{R})}(y_i)$-invariant deep neural networks, we can use the universal approximators defined in Yarotsky (2022, Definition 3.2). By this $\mathcal{H}_{\mathrm{Stab}_{\mathrm{SO}_2(\mathbb{R})}(y_i)\text{-}inv}$ and Theorem 2, we obtain universal approximators for $\mathrm{SO}_2(\mathbb{R})$-equivariant maps.

### 4.4 The translation group $\mathbb{T}$ for $\mathbb{R}^2$

It is known that the convolutional layers of convolution neural networks are translation equivariant. In this subsection, we consider an example for $X = Y = \mathbb{R}^2$ and the actions of the translation group $\mathbb{T}$ for $\mathbb{R}^2$ on $V^X$ and $W^Y$. The translation group $\mathbb{T}$ is defined by

$$\mathbb{T} = \{T_{\boldsymbol{v}} \mid \boldsymbol{v} \in \mathbb{R}^2\},$$

where each translation $T_{\boldsymbol{v}}$ is defined by $T_{\boldsymbol{v}}(\boldsymbol{x}) = \boldsymbol{x} + \boldsymbol{v}$. For $f \in V^X$, the action of $T_{\boldsymbol{v}}$ on $f$ is defined by

$$(T_{\boldsymbol{v}} \cdot f)(\boldsymbol{x}) = f(T_{\boldsymbol{v}}^{-1}(\boldsymbol{x})) = f(\boldsymbol{x} - \boldsymbol{v}).$$

We consider an $\mathbb{T}$-equivariant (i.e., translation equivariant) map $F \colon V^{\mathbb{R}^2} \to W^{\mathbb{R}^2}$. In this case, for $(0, 0) \in Y = \mathbb{R}^2$, the orbit $O_{(0,0)} = \mathbb{T} \cdot (0, 0)$ is same as $Y = \mathbb{R}^2$ and $\mathrm{Stab}_{\mathbb{T}}((0, 0)) = \{\mathrm{Id}\}$. Then, by Theorem 1, the $\mathbb{T}$-equivariant map $F$ can be written by

$$F(f) = (\widetilde{F}_{(0,0)}(T_{\boldsymbol{v}} \cdot f))_{\boldsymbol{v} \in Y}.$$

In particular, $\mathbb{T}$-equivariant map $F$ is determined by one $\{\mathrm{Id}\}$-invariant map (i.e., usual map without invariance) $\widetilde{F}_{(0,0)} \colon V^X \to W$. In this situation, by Theorem 2, if there is a universal approximators $\widehat{\widetilde{F}}_{(0,0)}$ of $\widetilde{F}_{(0,0)}$, $\Psi(\widehat{\widetilde{F}}_{(0,0)})$ approximates the map $F = \Psi(\widetilde{F}_{(0,0)})$.

## 5 Application to the numbers of parameters

### 5.1 A relation of the numbers of parameters for invariant/equivariant maps

In the case of finite group $G$, we investigate the number of parameters from the point of view of approximations. Let $G$ be a group and $X, Y$ be two finite $G$-sets. Let $Y = \bigsqcup_{i \in I} O_{y_i}$ be the $G$-orbit decomposition of $Y$ and $K \subset V^X$ be a compact subset which is stable by the $G$-action. For $F \in \mathrm{Equiv}_G^{\mathrm{cont}}(K, W^Y)$, we define $c_G^{\mathrm{eq}}(\varepsilon; F)$ to be the minimum number of the parameters of $G$-equivariant deep neural networks that can approximate $F$ in accuracy $\varepsilon > 0$. More precisely, there exists a $G$-equivariant deep neural network $\widehat{F}$ of which the number of parameters is equal to $c_G^{\mathrm{eq}}(\varepsilon; F)$ and $\sup_{f \in K} \|\widehat{F}(f) - F(f)\|_{W^Y} \leq \varepsilon$, and there is no such $G$-equivariant deep neural network of which the number of parameters is less than $c_G^{\mathrm{eq}}(\varepsilon; F)$. Similarly, for subgroup $H$ of $G$ and $\widetilde{F} \in \mathrm{Inv}_H^{\mathrm{cont}}(K, W)$, we define $c_H^{\mathrm{inv}}(\varepsilon; \widetilde{F})$ to be the minimum number of the parameters of $H$-invariant deep neural networks that can approximate $\widetilde{F}$ in accuracy $\varepsilon > 0$. The existences of such $c_H^{\mathrm{inv}}(\varepsilon; \widetilde{F})$ and $c_G^{\mathrm{eq}}(\varepsilon; F)$ are guaranteed by the results for universal approximation theorems such as Maron et al. (2019b) or Theorem 2 in this volume.

For $F \in \mathrm{Equiv}_G^{\mathrm{cont}}(K, W^Y)$, $\widetilde{F}_i \in \mathrm{Inv}_{\mathrm{Stab}_G(y_i)}(V^X, W)$ is defined by $(\widetilde{F}_i)_{i \in I} = \Phi(F)$ in the sense of Theorem 1. Then, we have the following:

**Theorem 3.** *For any $\varepsilon > 0$, the following holds:*

$$\max_{i \in I} c_{\mathrm{Stab}_G(y_i)}^{\mathrm{inv}}(\varepsilon; \widetilde{F}_i) \leq c_G^{\mathrm{eq}}(\varepsilon; F) \leq \sum_{i \in I} c_{\mathrm{Stab}_G(y_i)}^{\mathrm{inv}}(\varepsilon; \widetilde{F}_i). \tag{14}$$

*Proof.* We first show the left inequality. Let $\widetilde{F}_{i_0} \in \mathrm{Inv}_{\mathrm{Stab}_G(y_{i_0})}(V^X, W)$ for an $i_0 \in I$. We can find a $G$-equivariant deep neural network $\widehat{F}$ such that $\|\widehat{F}(f) - F(f)\|_{W^Y} \leq \varepsilon$ for any $f \in K$ and the number of parameters of $\widehat{F}$ is equal to $c_G^{\mathrm{eq}}(\varepsilon; F)$. The existence of $\widehat{F}$ is guaranteed by Theorem 2. For this $F$, $\widehat{F}(\cdot)(y_{i_0})$ satisfies $\|\widehat{F}(f)(y_{i_0}) - \widetilde{F}_{i_0}(f)\|_W \leq \varepsilon$ and the number of parameters of the deep neural network $\widehat{F}(\cdot)(y_{i_0})$ is less than or equal to $c_G^{\mathrm{eq}}(\varepsilon; F)$. Thus, $\widetilde{F}_{i_0}$ can be approximated in accuracy $\varepsilon$ by a $\mathrm{Stab}_G(y_{i_0})$-invariant deep neural network of which the number of parameters is less than or equal to $c_G^{\mathrm{eq}}(\varepsilon; F)$. This implies $c_{\mathrm{Stab}_G(y_{i_0})}^{\mathrm{inv}}(\varepsilon; \widetilde{F}_{i_0}) \leq c_G^{\mathrm{eq}}(\varepsilon; F)$. Because $i_0 \in I$ is arbitrary, we have the first inequality of equation (14).

We next show the second inequality of equation (14). For any $i \in I$, there is a deep neural network $\widehat{\widetilde{F}}_i$ such that $\|\widehat{\widetilde{F}}_i(f) - \widetilde{F}_i(f)\|_W \leq \varepsilon$ for $f \in K$ and the number of parameters of this is $c_{\mathrm{Stab}_G(y_i)}^{\mathrm{inv}}(\varepsilon; \widetilde{F}_i)$. We set $\widehat{F} = \Phi((\widehat{\widetilde{F}}_i)_{i \in I})$. Then, by Theorem 1 and inequality equation (12), we have

$$
\begin{aligned}
\sup_{f \in K} \|\widehat{F}(f) - F(f)\|_{W^Y} &= \sup_{f \in K} \sup_{y \in Y} \|\widehat{F}(f)(y) - F(f)(y)\|_W \\
&= \sup_{f \in K} \left( \sup_{i \in I} \left( \sup_{\sigma \in G} \|\widehat{F}(f)(\sigma^{-1} \cdot y_i) - F(f)(\sigma^{-1} \cdot y_i)\|_W \right) \right) \\
&= \sup_{f \in K} \left( \sup_{i \in I} \left( \sup_{\sigma \in G} \|\widehat{F}(\sigma \cdot f)(y_i) - F(\sigma \cdot f)(y_i)\|_W \right) \right) \\
&\leq \sup_{i \in I} \left( \sup_{f \in K} \|\widehat{F}(f)(y_i) - F(f)(y_i)\|_W \right) \\
&= \sup_{i \in I} \left( \sup_{f \in K} \|\widehat{\widetilde{F}}_i(f) - \widetilde{F}_i(f)\|_W \right) \leq \varepsilon. \tag{15}
\end{aligned}
$$

Because the number of parameters of $\widehat{F}(f)$ is less than or equal to $\sum_{i \in I} c_{\mathrm{Stab}_G(y_i)}^{\mathrm{inv}}(\varepsilon; \widetilde{F}_i)$, any elements in $\mathrm{Equiv}_G^{\mathrm{cont}}(K, W^Y)$ can be approximated in accuracy $\varepsilon$ by a $G$-equivariant deep neural network of which the number of parameters is less than or equal to $\sum_{i \in I} c_{\mathrm{Stab}_G(y_i)}^{\mathrm{inv}}(\varepsilon; \widetilde{F}_i)$. This implies $c_G^{\mathrm{eq}}(\varepsilon; F) \leq \sum_{i \in I} c_{\mathrm{Stab}_G(y_i)}^{\mathrm{inv}}(\varepsilon; \widetilde{F}_i)$. This is the second inequality of equation (14). $\square$

## 5.2 Approximation rate for $G$-equivariant deep neural networks

In the case of $G = S_n$, $X = \{1, 2, \ldots, n\}$, $V = W = \mathbb{R}$, and $K = [0,1]^n \subset V^n$, Sannai et al. (2021, Theorem 4) showed an approximation rate of $S_n$-invariant ReLU deep neural networks for elements in a Hölder space. In this subsection, we generalize this result to $G$-invariant maps for a finite group $G \subset S_n$ and show an approximation rate for $G$-equivariant ReLU deep neural networks.

Let $\alpha$ be a positive constant. The Hölder space of order $\alpha$ is the space of continuous functions $f$ such that all the partial derivatives of $f$ up to order $\lfloor \alpha \rfloor$ exist and the partial derivatives of order $\lfloor \alpha \rfloor$ are $\alpha - \lfloor \alpha \rfloor$ Hölder-smooth. This space is usually denoted as $\mathcal{C}^{\alpha, \alpha - \lfloor \alpha \rfloor}$, but we denote it as $\mathcal{C}^\alpha$ here for convenience. For $f \colon K \to \mathbb{R}$, the Hölder norm is defined by

$$\|f\|_{\mathcal{C}^\alpha} := \max_{\beta : |\beta| < \lfloor \alpha \rfloor} \|\partial^\beta f(x)\|_{L^\infty(K)} + \max_{\beta : |\beta| = \lfloor \alpha \rfloor} \sup_{x, x' \in K, x \neq x'} \frac{|\partial^\beta f(x) - \partial^\beta f(x')|}{\|x - x'\|_\infty^{\alpha - \lfloor \alpha \rfloor}}.$$

For $B > 0$, a $B$-radius ball in the Hölder space on $K$ is defined as $\mathcal{C}_B^\alpha = \{f \in \mathcal{C}^\alpha \mid \|f\|_{\mathcal{C}^\alpha} \leq B\}$.

The following is a generalization of Sannai et al. (2021, Theorem 4) to finite group $G \subset S_n$.

**Theorem 4.** *Let $G \subset S_n$ be a finite group acting on the sets $X = Y = \{1, 2, \ldots, n\}$ and $Y = \bigsqcup_{i=1}^k O_i$ be the $G$-orbit decomposition, and $K = [0,1]^n \subset \mathbb{R}^n$. For any $\varepsilon > 0$, let $\mathcal{H}_{G\text{-inv}}$ be a set of $G$-invariant ReLU deep neural networks from $K$ to $\mathbb{R}$ which has at most $O(\log(1/\varepsilon))$ layers and $O(\varepsilon^{-n/\alpha} \log(1/\varepsilon))$ non-zero parameters. Then, for any $G$-invariant function $\widetilde{F} \in \mathcal{C}_B^\alpha$, there exists $\widehat{\widetilde{F}} \in \mathcal{H}_{G\text{-inv}}$ such that*

$$\sup_{f \in K} |\widehat{\widetilde{F}}(f) - \widetilde{F}(f)| \leq \varepsilon.$$

*Proof.* To generalize Sannai et al. (2021, Theorem 4), it is sufficient to define the fundamental domain $\Delta_G$ for finite group $G \subset S_n$ and a sorting map $\mathrm{Sort} \colon [0,1]^n \to \Delta_G$, and to show that $\mathrm{Sort}$ can be realized by ReLU deep neural networks. The remaining part of the proof is similar to the proofs of Sannai et al. (2021, Proposition 9, Theorem 4)

Let $\{1, 2, \ldots, n\} = \bigsqcup_{i=1}^k O_i$ be a $G$-orbit decomposition and $I_j = \{i_1^j, \ldots, i_{m_j}^j\}$ satisfying $i_\ell^j < i_{\ell'}^j$ for $\ell < \ell'$.

Then, a fundamental domain $\Delta_G$ for $G$ is given as

$$\Delta_G = \{(x_1, \ldots, x_n)^\top \in [0,1]^n \mid x_{i_\ell^j} \geq x_{i_{\ell'}^j} \text{ if } \ell < \ell' \text{ for } j = 1, 2, \ldots, k\}.$$

We consider the reordered index $(i_1^1, i_2^1, \ldots, i_{m_1}^1, i_1^2, \ldots, i_{m_2}^2, \ldots, i_1^k, \ldots, i_{m_k}^k)$. We define the permutation $\sigma_0$ from $(1, 2, \ldots, n)$ to this index as

$$\sigma_0 = \begin{pmatrix} 1 & 2 & \cdots & n \\ i_1^1 & i_2^1 & \cdots & i_{m_k}^k \end{pmatrix}.$$

Then, by $\sigma_0^{-1}$, $(x_1, x_2, \ldots x_n)^\top \in [0,1]^n$ is permutated as

$$\sigma_0^{-1} \cdot (x_1, x_2, \ldots, x_n)^\top = (x_{\sigma_0(1)}, x_{\sigma_0(2)}, \ldots, x_{\sigma_0(n)})^\top = (x_{i_1^1}, x_{i_2^1}, \ldots, x_{i_{m_k}^k})^\top.$$

We remark that this permutation can be realized by a linear map. On the other hand, we define $\mathrm{Sort}_G \colon [0,1]^n \to [0,1]^n$ by

$$\mathrm{Sort}_G(x_1, \ldots, x_n) =$$
$$(\max_1^1(x_1, \ldots, x_n), \max_2^1(x_1, \ldots, x_n), \ldots, \max_{m_k}^k(x_1, \ldots, x_n)).$$

Here, $\max_\ell^j(x_1, \ldots, x_n)$ is the $\ell$-th largest value in $\{x_{p_{j-1}+1}, x_{p_{j-1}+2} \ldots, x_{p_j}\}$, where $p_j = \sum_{i=1}^{j-1} m_i$. As proven in Sannai et al. (2021, Proposition 8), we can show that the map $\mathrm{Sort}_G$ is represented by deep

neural networks with ReLU activation functions. We define $\widetilde{\text{Sort}}_G = \sigma_0 \circ \text{Sort}_G \circ \sigma_0^{-1}$. Then, the image $\widetilde{\text{Sort}}_G(x_1, \ldots, x_n)$ is in the fundamental domain $\Delta_G$ and $\widetilde{\text{Sort}}_G$ can also be realized by a ReLU deep neural network.

By applying Sannai et al. (2021, Proposition 9 and Proof of Theorem 4) with this $\widetilde{\text{Sort}}_G$, we can complete the proof. □

As mentioned in Sannai et al. (2021) for $G = S_n$, also for finite group $G \subset S_n$, the approximation rate in Theorem 4 is the optimal rate without invariance obtained by Yarotsky (2022, Theorem 1). Thus, we show that $G$-invariant deep neural networks can also achieve the optimal approximation rate even with invariance.

By combining Theorem 4 and similar argument in the proof of Theorem 3, we can show the approximation rate for $G$-equivariant maps:

**Corollary 1.** *Let $G \subset S_n$ be a finite group acting on the sets $X = Y = \{1, 2, \ldots, n\}$ and $Y = \bigsqcup_{i=1}^k O_i$ be the $G$-orbit decomposition. Let $V = W = \mathbb{R}$ and $K = [0,1]^n \subset \mathbb{R}^n$. For any $\varepsilon > 0$, let $\mathcal{H}_{G-equiv}$ be a set of $G$-equivariant ReLU deep neural networks from $K$ to $\mathbb{R}^n$ with at most $O(\log(1/\varepsilon))$ layers and $O(k\varepsilon^{-n/\alpha} \log(1/\varepsilon))$ non-zero parameters. Then, for any $G$-equivariant map $F \in (\mathcal{C}_B^\alpha)^n$, there exists $\widehat{F} \in \mathcal{H}_{G-equiv}$ such that*

$$\sup_{f \in K} \|\widehat{F}(f) - F(f)\|_{\mathbb{R}^n} \leq \varepsilon.$$

*Proof.* Let $F \in (\mathcal{C}_B^\alpha)^n$ be a $G$-equivariant map and $(\widetilde{F}_i)_{i \in I} = \Phi(F)$. Then, by Theorem 1, $\widetilde{F}_i$ is a $\text{Stab}_G(y_i)$-invariant continuous function on $K$. By Theorem 4, there is an element $\widehat{\widetilde{F}}_i$ in $\mathcal{H}_{\text{Stab}_G(y_i)\text{-inv}}$ such that

$$\sup_{f \in K} |\widehat{\widetilde{F}}_i(f) - \widetilde{F}_i(f)| \leq \varepsilon.$$

We set $\widehat{F} = \Psi((\widehat{\widetilde{F}}_i)_{i \in I})$. Then, by the similar argument in inequality equation (15), we have

$$\sup_{f \in K} \|\widehat{F}(f) - F(f)\|_{\mathbb{R}^n} \leq \sup_{i \in I} \sup_{f \in K} |\widehat{\widetilde{F}}_i(f) - \widetilde{F}_i(f)| \leq \varepsilon.$$

Now, $\widehat{F}$ has at most $O(\log(1/\varepsilon)$ layers and $O(k\varepsilon^{-n/\alpha} \log(1/\varepsilon))$ non-zero parameters because of the assumptions for $\widehat{\widetilde{F}}_i$ $(i = 1, 2, \ldots, k)$. This concludes the proof. □

# 6 Conclusion

In this study, we explained a fundamental relationship between equivariant maps and invariant maps for stabilizer subgroups. The relation allows any equivariant tasks for a group to be reduced to some invariant tasks for some stabilizer subgroups. As an application of this relation, we proposed the construction of universal approximators for equivariant continuous maps by using some invariant deep neural networks.

Although our model is different from the other standard models introduced by Zaheer et al. (2017) or Maron et al. (2019b), the number of parameters of our models is fewer than fully connected models. Moreover, by the relation, we showed inequalities of the number of parameters of invariant or equivariant deep neural networks to approximate any continuous invariant/equivariant maps. We also showed an approximation rate of $G$-equivariant ReLU deep neural networks for elements of a Hölder space for finite group $G$.

The invariant-equivariant relation is fundamental and applicable to a wide variety of situations. We believe that this study will lead to further research in the field of machine learning.

**Acknowledgments**

AT was supported in part by JSPS KAKENHI Grant No. JP20K03743, JP23H04484 and JST PRESTO JPMJPR2123. YT was partly supported by JSPS KAKENHI Grant Numbers JP21K11763 and Grant for

Basic Science Research Projects from The Sumitomo Foundation Grant Number 200484. MC was partly supported by the French National Research Agency in the framework of the LabEx PERSYVAL-Lab (ANR-11-LABX-0025-01). The previous version of this work was done during MC's visit to RIKEN AIP.

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

## A  Groups, actions, and orbits

To introduce a precise statement of the main theorem, we briefly review several notions of groups, actions, and orbits. In general, a group is a set of transformations and an action of the group on a set is a transformation rule. This is further explained in the following examples. For more details, we refer the readers to Rotman (2012).

Let $G$ be a set with a product $\sigma\tau \in G$ for any elements $\sigma, \tau$ of $G$. Then, $G$ is called a group if $G$ satisfies the following conditions:

1. (Existence of the unit) There is an element $\epsilon \in G$ such that $\epsilon\sigma = \sigma\epsilon = x$ for all $\sigma \in G$.

2. (Existence of the inverse element) For any $\sigma \in G$, there is an element $\sigma^{-1} \in G$ such that $\sigma\sigma^{-1} = \sigma^{-1}\sigma = \epsilon$.

3. (Associativity) For any $\sigma, \tau, \phi \in G$, $(\sigma\tau)\phi = \sigma(\tau\phi)$.

We call $G$ a finite group if the number of elements of $G$ is finite. If the product of $G$ is commutative, i.e., for any $\sigma, \tau \in G$, $\sigma\tau = \tau\sigma$ holds, then $G$ is called an abelian (or a commutative) group. Let $G$ be a group and $H$ a subset of $G$. We call $H$ a subgroup of $G$ if $H$ is a group with the same product as that of $G$.

**Example 4.** *(i)  The permutation group $S_n$ is one of the most important examples of finite groups:*

$$S_n = \{\sigma \colon \{1, \ldots, n\} \to \{1, \ldots, n\} \mid \text{bijective}\}$$

*and the product of $\sigma, \tau \in S_n$ is given by the composition $\sigma \circ \tau$ as maps. The permutation group $S_n$ is also called by the symmetric group.*

*(ii) The special orthogonal group $\mathrm{SO}_2(\mathbb{R})$ is also an abelian group:*

$$\mathrm{SO}_2(\mathbb{R}) = \{A \in \mathbb{R}^{2\times 2} \mid A^\top A = I, \ \det A = 1\}$$
$$= \left\{ \begin{pmatrix} \cos\theta & \sin\theta \\ -\sin\theta & \cos\theta \end{pmatrix} \ \middle| \ \theta \in [0, 2\pi) \right\},$$

*where $I$ is the $2 \times 2$ unit matrix. As seen above, the elements of $\mathrm{SO}_2(\mathbb{R})$ include the rotations on $\mathbb{R}^2$.*

Next, we review group actions on sets. Let $X$ be a set. A left (resp. right) action of $G$ on $X$ is defined as a map $X \to X; x \mapsto \sigma \cdot x$ for $\sigma \in G$ satisfying the following:

1. For any $x \in X$, $\epsilon \cdot x = x$.

2. For any $\sigma, \tau \in G$ and $x \in X$, $(\sigma\tau) \cdot x = \sigma \cdot (\tau \cdot x)$ (resp. $(\sigma\tau) \cdot x = \tau \cdot (\sigma \cdot x)$).

Then, we say that $G$ acts on $X$ from left (resp. right). Here, the reason why we call the latter a right action is that the condition $(\sigma\tau) \cdot x = \tau \cdot (\sigma \cdot x)$ seems as if the condition "$x \cdot (\sigma\tau) = (x \cdot \sigma) \cdot \tau$".

**Example 5.** *(i) The permutation group $S_n$ acts on the set $\{1, 2, \ldots, n\}$ from left by the permutation $\sigma \cdot i = \sigma(i)$.*

*(ii) The special orthogonal group $\mathrm{SO}_2(\mathbb{R})$ acts on $\mathbb{R}^2$ from left by the rotation transformations.*

We introduce a subgroup that is often considered in this article. Let $G$ be a group acting on a set $X$ from left. For an element $x \in X$, the subset $\text{Stab}_G(x)$ of $G$ consisting of elements stabilizing $x$ is a subgroup of $G$:

$$\text{Stab}_G(x) = \{\sigma \in G \mid \sigma \cdot x = x\}.$$

This group $\text{Stab}_G(x)$ is called the stabilizer subgroup of $G$ with respect to $x$.

Next, we introduce the notion of orbits by the group actions. An orbit is the set of elements obtained by transformations of a fixed base element. Let $G$ be a group acting on $X$ from left. Then, for $x \in X$, we define the $G$-orbit $O_x$ of $x$ as

$$O_x = G \cdot x = \{\sigma \cdot x \mid \sigma \in G\}.$$

Then, it is easy to demonstrate that $X$ can be decomposed into a disjoint union of the $G$-orbits $X = \bigsqcup_{i \in I} O_{x_i}$. We call this the $G$-orbit decomposition of $X$. We remark that the index set $I$ might be infinite.

**Example 6.** *(i) We consider the permutation action of $S_n$ on $X = \{1, 2, \ldots, n\}$. Then, the orbit of $1 \in X$ is same as $X$, i.e., $X = O_1 = S_n \cdot 1$.*

*(ii) The orbit of $\boldsymbol{X} \in \mathbb{R}^2$ of the action of $\text{SO}_2(\mathbb{R})$ on $X = \mathbb{R}^2$ is same as*

$$O_{\boldsymbol{X}} = \begin{cases} \{(a, b) \in \mathbb{R}^2 \mid a^2 + b^2 = \|\boldsymbol{X}\|_2^2\} & \text{if } \boldsymbol{X} \neq \boldsymbol{0}, \\ \{\boldsymbol{0}\} & \text{if } \boldsymbol{X} = \boldsymbol{0}. \end{cases}$$

*When $\|\boldsymbol{X}\|_2 = r$, then we can represent the orbits as $O_{(r,0)} = O_{\boldsymbol{X}}$. Hence, the orbit decomposition of $X = \mathbb{R}^2$ is same as $X = \bigsqcup_{r \geq 0} O_{(r,0)}$.*

Let $H$ be a subgroup of a finite group $G$. Then, $H$ acts on $G$ from left and right by the product: For $\tau \in H$ and $\sigma \in G$, $\tau$ acts on $\sigma$ from left (resp. right) by $\sigma \mapsto \tau\sigma$ (resp. $\sigma \mapsto \sigma\tau$). For any $\sigma \in G$, we define the set

$$\sigma H = \{\sigma\tau \in G \mid \tau \in H\}.$$

This is nothing but the right $H$-orbit of $\sigma$ for the above right action of $H$ on $G$. This is called the left coset of $H$ with respect to $\sigma$. Because the left coset is identical to the $H$-orbit for the right action of $H$, we can decompose $G$ into a disjoint union of the left cosets as a $H$-orbit decomposition:

$$G = \bigsqcup_{i \in I} \sigma_i H.$$

We refer to this as the left coset decomposition of $G$ by $H$. We define $G/H$ as the set of the left cosets of $G$ by $H$:

$$G/H = \{\sigma_i H \mid i \in I\}.$$

The right cosets, the right coset decomposition, and the set of right cosets $H \backslash G$ are also defined similarly.

Then, a relation exists between an orbit and a set of cosets:

**Proposition 2.** *Let $G$ be a group acting on a set $X$ from left. For any $x \in X$, the map*

$$G/\text{Stab}_G(x) \to O_x; \ \sigma\text{Stab}_G(x) \to \sigma \cdot x$$

*is bijective. In particular, if $G/\text{Stab}_G(x) = \{\sigma_i \text{Stab}_G(x) \mid i \in I\}$, then the orbit of $x$ in $X$ can be written by $O_x = \{\sigma_i \cdot x \mid i \in I\}$.*

*Proof.* It is easy to check well-definedness and bijectivity. $\square$

**Example 7.** *(i) For $G = S_n$ acting on $\{1, 2, \ldots, n\}$, the $G$-orbit of $1$ was only one, i.e.,*

$$O_1 = \sigma \cdot 1 = \{1, 2, \ldots, n\}.$$

*Hence, by Proposition 2, the following map is bijective:*

$$S_n/\text{Stab}_{S_n}(1) \to \{1, 2, \ldots, n\}; \ \sigma\text{Stab}_{S_n}(1) \mapsto \sigma(1).$$

*(ii) For $G = \mathrm{SO}_2(\mathbb{R})$, the orbit could be represented by $O_{(r,0)} = \mathrm{SO}_2(\mathbb{R}) \cdot (r, 0)$ for $r \geq 0$. Then, the stabilizer subgroup of $\mathrm{SO}_2(\mathbb{R})$ for $(r, 0)$ is*

$$\mathrm{Stab}_{\mathrm{SO}_2(\mathbb{R})}((r, 0)) = \begin{cases} \{I\} & \text{if } r > 0, \\ \mathrm{SO}_2(\mathbb{R}) & \text{if } r = 0. \end{cases}$$

*By Proposition 2, the map*

$$\mathrm{SO}_2(\mathbb{R})/\mathrm{Stab}_{\mathrm{SO}_2(\mathbb{R})}((r, 0)) \to O_{(r,0)}; \ \sigma \mathrm{Stab}_{\mathrm{SO}_2(\mathbb{R})}((r, 0)) \mapsto \sigma \cdot (r, 0)$$

*is bijective for any $r \geq 0$. Indeed, this is compatible with Example 6 (ii).*

One reason for the importance of the permutation group $S_n$ is the following proposition:

**Proposition 3.** *Any finite group $G$ can be realized as a subgroup of $S_n$ for some positive integer $n$.*

*Proof.* Let $n := |G|$ and $G = \{\sigma_1, \sigma_2, \ldots, \sigma_n\}$. For any $\sigma \in G$, $\sigma \sigma_i = \sigma_j$ for some $j \in \{1, 2, \ldots, n\}$, because the product $\sigma \sigma_i$ is in $G$. We set $\tau_\sigma(i) = j$. Then, we define a map $\tau \colon G \to S_n; \sigma \mapsto \tau_\sigma$. This map is injective and preserves the product, i.e., $\tau_{\sigma\sigma'} = \tau_\sigma \tau_{\sigma'}$. Through this map, $G$ can be regarded as a subgroup of $S_n$. $\square$

## B  Theoretical explanation of Remark 1

To explain the architecture of our models, we introduce some notions of representation theory.

Let $G$ be a finite group, $V$ be a vector space over $\mathbb{R}$, and $\mathrm{GL}(V)$ be the group of the automorphisms of $V$. Here, an automorphism of $V$ is a linear bijection from $V$ to $V$. Then, a group homomorphism $\rho \colon G \to \mathrm{GL}(V)$ is called a (linear) representation of $G$ on $V$. Defining a representation $\rho \colon G \to \mathrm{GL}(V)$ is equivalent to defining an action of $G$ on $V$ by $\rho$. Then, it is also known that considering a representation $\rho \colon G \to \mathrm{GL}(V)$ is equivalent to considering an $\mathbb{R}[G]$-module $V$. Here, $\mathbb{R}[G]$ is the group algebra defined as

$$\mathbb{R}[G] = \left\{ \sum_{\sigma \in G} a_\sigma \sigma \ \middle| \ a_\sigma \in \mathbb{R} \right\}.$$

Moreover, for an algebra $A$ with unit 1, an $A$-module $M$ is an abelian group with summation $+$, endowed with a "scalar product" by $A$, denoted as $\cdot$, satisfying

$$\sigma \cdot (x + x') = \sigma \cdot x + \sigma \cdot x',$$
$$(\sigma + \tau) \cdot x = \sigma \cdot x + \tau \cdot x,$$
$$(\sigma\tau) \cdot x = \sigma \cdot (\tau \cdot x),$$
$$1 \cdot x = x$$

for $\sigma, \tau \in A$ and $x, x' \in M$. This means that an $A$-module is an analog of a vector space with the scalar product by algebra $A$. When a representation $\rho \colon G \to \mathrm{GL}(V)$ is given, $V$ can be regarded as an $\mathbb{R}[G]$-module by the action

$$\left( \sum_{\sigma \in G} a_\sigma \sigma \right) \cdot v = \sum_{\sigma \in G} a_\sigma \rho(\sigma)(v).$$

for $\sum_{\sigma \in G} a_\sigma \sigma \in \mathbb{R}[G]$ and $v \in V$.

Two representations $(\rho, V), (\rho', V')$ of a group $G$ are called isomorphic, denoted $(\rho, V) \simeq (\rho', V')$ if there is a linear isomorphism $\Xi \colon V \to V'$ satisfying the commutativity

$$\Xi(\rho(\sigma)(v)) = \rho'(\sigma)(\Xi(v)) \tag{16}$$

for any $\sigma \in G$ and $v \in V$. A linear map (not necessarily isomorphism) satisfying commutativity equation (16) is called an intertwining operator. We set $\mathrm{Hom}_G(V, V')$ as the set of intertwining operators from $(\rho, V)$ to $(\rho', V')$.

For a subgroup $H \subset G$ and a representation $\rho\colon G \to \mathrm{GL}(V)$, the restriction map $\rho|_H\colon H \to \mathrm{GL}(V)$ is a representation of $H$ on $V$. This representation is called the restricted representation of $\rho$ to $H$ and denoted by $\mathrm{Res}_H^G(\rho)$. Then, considering the restricted representation $\rho|_H$ is equivalent to considering $V$ as an $\mathbb{R}[H]$-module.

On the other hand, for a subgroup $H \subset G$ and a representation $\rho'\colon H \to \mathrm{GL}(W)$ of $H$ on $W$, the coefficient extension of $\mathbb{R}[H]$-module $W$ to $\mathbb{R}[G]$

$$\mathbb{R}[G] \otimes_{\mathbb{R}[H]} W$$

is an $\mathbb{R}[G]$-module. Here, the tensor product $\mathbb{R}[G] \otimes_{\mathbb{R}[H]} W$ is defined as the quotient $(\mathbb{R}[G] \times W)/\sim$ of the direct product $\mathbb{R}[G] \times W$, by the equivalent relation $\sim$ which is defined for $\sigma, \sigma' \in \mathbb{R}[G]$, $\tau \in H$, $w, w' \in W$ and $c \in \mathbb{R}$ by:

$$(\sigma + \sigma', w) \sim (\sigma, w) + (\sigma', w),$$
$$(c\sigma, w) \sim (\sigma, cw),$$
$$(\sigma, w + w') \sim (\sigma, w) + (\sigma, w'),$$
$$(\sigma\tau, w) \sim (\sigma, \rho'(\tau)(w)).$$

Then, the action of $\sigma' \in G$ on an element $\sigma \otimes w \in \mathbb{R}[G] \otimes_{\mathbb{R}[H]} W$ is defined by

$$\sigma'(\sigma \otimes w) = (\sigma'\sigma) \otimes w.$$

We call the representation defined by this the induced representation of $\rho'$ to $G$ and denote by $\mathrm{Ind}_H^G(\rho')$. We remark that the following equality holds:

$$\sigma\tau \otimes w = \sigma \otimes \rho'(\tau)(w)$$

for any $\sigma \in G$, $\tau \in H$, $w \in W$. Let $G = \bigsqcup_{i=1}^m H\sigma_i$ be the right coset decomposition of $G$ by $H$. Then, we have the directed sum decomposition

$$\mathbb{R}[G] = \bigoplus_{i=1}^m \sigma_i^{-1} \mathbb{R}[H]. \tag{17}$$

By equation (17), we have

$$\mathbb{R}[G] \otimes_{\mathbb{R}[H]} W = \left( \bigoplus_{i=1}^m \sigma_i^{-1} \mathbb{R}[H] \right) \otimes_{\mathbb{R}[H]} W = \bigoplus_{i=1}^m \sigma_i^{-1} \otimes_{\mathbb{R}[H]} W \simeq W^m$$

Therefore, the dimension of $\mathbb{R}[G] \otimes_{\mathbb{R}[H]} W$ is $m \dim W = (G : H) \dim W$, where $(G : H)$ is the index of $H$ in $G$.

Let $G \subset S_n$ be a finite group, $F\colon V^n \to W^n$ be a $G$-equivariant continuous map, and let $Y = \{1, 2, \ldots, n\} = \bigsqcup_{i=1}^m O_{y_i}$ be a $G$-orbit decomposition with $O_{y_i} = \{\sigma^{-1}(y_i) \mid \sigma \in G\}$ of $y_i \in Y$. By Theorem 1, we have $\Phi(F) = (\widetilde{F}_i)_{i=1}^m$ for the $\mathrm{Stab}_G(y_i)$-invariant continuous map $\widetilde{F}_i(\cdot) = F(\cdot)(y_i)$. Then, we have an approximator $\widehat{\widetilde{F}}_i\colon V^n \to W$ as a $\mathrm{Stab}_G(y_i)$-invariant deep neural network of the $\mathrm{Stab}_G(y_i)$-invariant map $\widetilde{F}_i$ (cf. Segol & Lipman (2019), Ravanbakhsh (2020)) and, by Theorem 2, can reconstruct an approximator $\widehat{F} = \Psi((\widehat{\widetilde{F}}_i)_{i=1}^m)$.

Let $G = \bigsqcup_{j=1}^{\ell_i} \mathrm{Stab}_G(y_i)\sigma_{ij}$ be a right coset decomposition for $\mathrm{Stab}_G(y_i)$. Then, by Proposition 2, we have $O_{y_i} = \{\sigma_{ij}^{-1}(y_i) \mid j = 1, \ldots, \ell_i\}$. By using this, an approximator $\widehat{F}$ of $F$ is constructed from $(\widehat{\widetilde{F}}_i)_{i=1}^m$ by

$$\widehat{F}(\boldsymbol{X}) = ((\widehat{\widetilde{F}}_i(\sigma_{ij} \cdot \boldsymbol{X})_{j=1}^{\ell_i})_{i=1}^m \tag{18}$$

for $\boldsymbol{X} = (\boldsymbol{x}_1, \ldots, \boldsymbol{x}_n) \in V^n$.

We focus on a $\text{Stab}_G(y_i)$-invariant approximator $\widehat{\widetilde{F}}_i \colon V^n \to W$. By Segol & Lipman (2019), we assume that $\widehat{\widetilde{F}}_i$ can be realized as a deep neural network model as

$$V^n \xrightarrow{\varphi_1^{(i)}} (V_1^{(i)})^n \xrightarrow{\varphi_2^{(i)}} (V_2^{(i)})^n \xrightarrow{\varphi_3^{(i)}} \dots \xrightarrow{\varphi_d^{(i)}} (V_d^{(i)})^n \xrightarrow{\varphi_{d+1}^{(i)}} (V_{d+1}^{(i)})^n = W^n,$$

where $V_k^{(i)}$ is a finite dimensional vector space over $\mathbb{R}$ and the $\varphi_k^{(i)}$ is a map defined by

$$\varphi_k^{(i)}(\boldsymbol{X}) = \text{ReLU}(W_k^{(i)}\boldsymbol{X} + \boldsymbol{B}_k^{(i)})$$

for $\boldsymbol{X} \in (V_{k-1}^{(i)})^n$, $W_k^{(i)} \in \mathbb{R}^{n \dim V_k^{(i)} \times n \dim V_{k-1}^{(i)}}$, and $\boldsymbol{B}_k^{(i)} = (\boldsymbol{b}_{k1}^{(i)}, \dots, \boldsymbol{b}_{kn}^{(i)}) \in (V_k^{(i)})^n$ such that $W_k^{(i)}$ (resp. $\boldsymbol{B}_k^{(i)}$) is $\text{Stab}_G(y_i)$-equivariant (resp. $\text{Stab}_G(y_i)$-invariant), i.e., for any $\boldsymbol{X} \in (V_{k-1}^{(i)})^n$ and any $\sigma \in \text{Stab}_G(y_i)$,

$$W_k^{(i)}(\sigma \cdot \boldsymbol{X}) = \sigma \cdot (W_k^{(i)}\boldsymbol{X}), \quad \sigma \cdot \boldsymbol{B}_k^{(i)} = \boldsymbol{B}_k^{(i)}$$

holds. Here, $\text{Stab}_G(y_i) \subset G \subset S_n$ acts on $(V_k^{(i)})^n$ by the restriction of the permutation action.

By combining the equation (18) and this notation, the weight matrix from the input layer $V^n$ to the first hidden layer $\bigoplus_{i=1}^m \bigoplus_{j=1}^{\ell_i} (V_1^{(i)})^n$ of $\widehat{F}$ is

$$((W_1^{(i)}\sigma_{ij})_{j=1}^{\ell_i})_{i=1}^m.$$

Then, by the action of $\sigma \in G$, we have

$$\begin{aligned}
W_k^{(i)}\sigma_{ij}(\sigma \cdot \boldsymbol{X}) &= W_k^{(i)}((\sigma_{ij}\sigma) \cdot \boldsymbol{X}) \\
&= W_k^{(i)}((\tau\sigma_{ij'}) \cdot \boldsymbol{X}) \\
&= W_k^{(i)}\tau(\sigma_{ij'} \cdot \boldsymbol{X}) \\
&= \tau W_k^{(i)}(\sigma_{ij'} \cdot \boldsymbol{X}),
\end{aligned}$$

where $j' \in Y$ and $\tau \in \text{Stab}_G(y_i)$ are determined by $\sigma_{ij}\sigma = \tau\sigma_{ij'} \in \text{Stab}_G(y_i)\sigma_{ij'}$.

In particular, by the action of $\sigma \in G$, the $(i,j)$-th part $W_k^{(i)}\sigma_{ij}(\boldsymbol{X}) \in (V_k^{(i)})^n$ is replaced by $(i,j')$-th part $\tau W_k^{(i)}(\sigma_{ij'} \cdot \boldsymbol{X})$ with the action of $\tau = \sigma_{ij}\sigma\sigma_{ij'}^{-1} \in \text{Stab}_G(y_i)$. From this observation, we define the action of $G$ on the $i$-th part $\bigoplus_{j=1}^{\ell_i} (V_k^{(i)})^n$ as follows: For $\sigma \in G$ and $j = 1, 2, \dots, \ell_i$, we define $\phi_\sigma^{(i)}(j) \in \{1, 2, \dots, \ell_i\}$ such that

$$\sigma_{ij}\sigma = \tau\sigma_{i\phi_\sigma^{(i)}(j)} \in H_i\sigma_{i\phi_\sigma^{(i)}(j)}. \tag{19}$$

Then, for $\boldsymbol{V}^{(i)} = (\boldsymbol{v}_j^{(i)})_{j=1}^{\ell_i} \in \bigoplus_{j=1}^{\ell_i} (V_k^{(i)})^n \subset \bigoplus_{i=1}^m \bigoplus_{j=1}^{\ell_i} (V_k^{(i)})^n$, we define the action of $\sigma \in G$ by

$$\sigma * \boldsymbol{V}^{(i)} = ((\sigma_{ij}\sigma\sigma_{i\phi_\sigma^{(i)}(j)}^{-1}) \cdot \boldsymbol{v}_{\phi_\sigma^{(i)}(j)}^{(i)})_{j=1}^{\ell_i}. \tag{20}$$

The following theorem shows that this action $*$ of $G$ is equivalent to the induced representation of the restricted representation on $(V_k^{(i)})^n$.

**Theorem 5.** *For our proposed model, we define the action of $G$ on the $k$-th layer $\bigoplus_{i=1}^m \bigoplus_{j=1}^{\ell_i} (V_k^{(i)})^n$ by the action $*$ defined in equation (20). Then, the representation defined by the action $*$ is isomorphic to the sum of the induced representation of the restrictions*

$$\bigoplus_{i=1}^m \text{Ind}_{\text{Stab}_G(y_i)}^G(\text{Res}_{\text{Stab}_G(y_i)}^G((V_k^{(i)})^n)).$$

*Moreover, for any $k \geq 1$, any affine maps from $(k-1)$-th layer to $k$-th layer are $G$-equivariant for the action $*$. (Only for the input layer, the action is the restriction of the permutation action to $G$.)*

*Proof.* We set $H_i = \mathrm{Stab}_G(y_i)$ and $G = \bigsqcup_{j=1}^{\ell_i} H_i \sigma_{ij}$. For $i$-th part, the induced representation of the restricted representation of $(V_k^{(i)})^n$ can be regarded as the $\mathbb{R}[G]$-module

$$\mathbb{R}[G] \otimes_{\mathbb{R}[H_i]} (V_k^{(i)})^n = \bigoplus_{j=1}^{\ell_i} \sigma_{ij}^{-1} \otimes (V_k^{(i)})^n.$$

Then, by the action of $\sigma$, an element $\sigma_{ij}^{-1} \otimes \boldsymbol{v} \in \bigoplus_{j=1}^{\ell_i} \sigma_{ij}^{-1} \otimes (V_k^{(i)})^n$ is changed as

$$\begin{aligned}
\sigma \cdot (\sigma_{ij}^{-1} \otimes \boldsymbol{v}) &= (\sigma \sigma_{ij}^{-1}) \otimes \boldsymbol{v} = (\sigma_{ij} \sigma^{-1})^{-1} \otimes \boldsymbol{v} \\
&= (\tau \sigma_{ij'})^{-1} \otimes \boldsymbol{v} = \sigma_{ij'}^{-1} \tau^{-1} \otimes \boldsymbol{v} = \sigma_{ij'} \otimes (\tau^{-1} \cdot \boldsymbol{v}),
\end{aligned} \tag{21}$$

where $j' \in Y$ and $\tau \in H_i$ are determined by $\sigma_{ij}\sigma^{-1} = \tau\sigma_{ij'} \in H_i\sigma_{ij'}$. We remark that this $j'$ is equal to $\phi_{\sigma^{-1}}^{(i)}(j)$ by the notation defined in equation (19). This means that by the action of $\sigma \in G$, the $(i,j)$-th part $\sigma_{ij}^{-1} \otimes \boldsymbol{v}$ is replaced to the $(i, \phi_{\sigma^{-1}}^{(i)}(j))$-part $\sigma_{i\phi_{\sigma^{-1}}^{(i)}(j)} \otimes (\tau \cdot \boldsymbol{v})$ with the action of $\tau = \sigma_{ij}\sigma^{-1}\sigma_{i\phi_{\sigma^{-1}}^{(i)}(j)}^{-1} \in H_i$.

We define the map $\Xi$ from the $i$-th part of $k$-th hidden layer $\bigoplus_{j=1}^{\ell_i} (V_k^{(i)})^n$ to $\mathbb{R}[G] \otimes_{\mathbb{R}[H_i]} (V_k^{(i)})^n = \bigoplus_{j=1}^{\ell_i} \sigma_{ij}^{-1} \otimes (V_k^{(i)})^n$ by

$$\Xi \colon \boldsymbol{V}^{(i)} = (\boldsymbol{v}_j^{(i)})_{j=1}^{\ell_i} \longmapsto \sum_{j=1}^{\ell_i} \sigma_{ij}^{-1} \otimes \boldsymbol{v}_j^{(i)}.$$

By definition, this map is bijective and equivariant for the action $*$ of $\sigma \in G$ on $\bigoplus_{j=1}^{\ell_i} (V_k^{(i)})^n$ by equation (20) and on $\mathbb{R}[G] \otimes_{\mathbb{R}[H_i]} (V_k^{(i)})^n$ by equation (21). Indeed, we have

$$\begin{aligned}
\Xi(\sigma * \boldsymbol{V}^{(i)}) &= \Xi\big(((\sigma_{ij}\sigma\sigma_{i\phi_\sigma^{(i)}(j)}^{-1}) \cdot \boldsymbol{v}_{\phi_\sigma^{(i)}(j)}^{(i)})_{j=1}^{\ell_i}\big) \\
&= \sum_{j=1}^{\ell_i} \sigma_{ij}^{-1} \otimes (\sigma_{ij}\sigma\sigma_{i\phi_\sigma^{(i)}(j)}^{-1}) \cdot \boldsymbol{v}_{\phi_\sigma^{(i)}(j)}^{(i)} \\
&= \sum_{j=1}^{\ell_i} \sigma_{ij}^{-1}(\sigma_{ij}\sigma\sigma_{i\phi_\sigma^{(i)}(j)}^{-1}) \otimes \boldsymbol{v}_{\phi_\sigma^{(i)}(j)}^{(i)} \\
&= \sum_{j=1}^{\ell_i} (\sigma\sigma_{i\phi_\sigma^{(i)}(j)}^{-1}) \otimes \boldsymbol{v}_{\phi_\sigma^{(i)}(j)}^{(i)} \\
&= \sigma \sum_{j=1}^{\ell_i} \sigma_{i\phi_\sigma^{(i)}(j)}^{-1} \otimes \boldsymbol{v}_{\phi_\sigma^{(i)}(j)}^{(i)} \\
&= \sigma \sum_{j=1}^{\ell_i} \sigma_{ij}^{-1} \otimes \boldsymbol{v}_j^{(i)} = \sigma \cdot \Xi(\boldsymbol{V}^{(i)}).
\end{aligned}$$

This means that $\bigoplus_{j=1}^{\ell_i} (V_k^{(i)})^n$ and $\mathbb{R}[G] \otimes_{\mathbb{R}[H_i]} (V_k^{(i)})^n$ are isomorphic as $\mathbb{R}[G]$-modules.

The equivariance of affine maps is deduced by the definition of the action $*$. This concludes the proof. $\square$

## C  Theoretical explanation of Remark 2

In the remaining part, we argue about the number of parameters of the affine maps between layers. If a weight matrix between two layers of neural networks is equivariant by some $G$-action, this can be regarded as an intertwining operator. To count the number of parameters of intertwining operators, we need the notion of irreducible representations. We here review these notions briefly.

For a representation $(\rho, V)$ of group $G$, we call a subspace $W \subset V$ $G$-invariant subspace if $\rho(\sigma)(W) \subset W$ for any $\sigma \in G$. Then, $(\rho, W)$ is also a representation of $G$. This representation $(\rho, W)$ is called a subrepresentation of $(\rho, V)$. When a representation $(\rho, V)$ has no nontrivial subrepresentation, this is called irreducible representation. This means that if $(\rho, W)$ is a subrepresentation of the irreducible representation $(\rho, V)$, then $(\rho, W)$ is equal to $(\rho, \{0\})$ or $(\rho, V)$. By Maschke's theorem (Curtis & Reiner, 1966, (10.7),(10.8)), any finite-dimensional representation over $\mathbb{R}$ can be factorized to the direct sum of irreducible representations up to isomorphic. In particular, irreducible representations are "building blocks" of representations.

By Schur's Lemma (Curtis & Reiner, 1966, (27.3)), for two irreducible representations $(\rho, V), (\rho'V')$, $\mathrm{Hom}_G(V, V') = 0$ holds if and only if $(\rho, V)$ is not isomorphic to $(\rho', V')$. Thus, by this fact with Maschke's theorem, any intertwining operator can be factorized to the sum of intertwining operators between irreducible representations.

For $G = S_n$, by the argument with Young's diagrams, we can calculate how the irreducible representations change by induction and restriction of them. By combining such argument and Theorem 5, we can calculate the number of parameters of the weight matrix between the input layer and the first hidden layer as follows:

**Proposition 4.** *For $G = S_n$, $S_n$-equivariant continuous map $F\colon V^n \to W^n$ can be recovered by one $\mathrm{Stab}_{S_n}(1)$-invariant map $\widetilde{F}_1$ as $F = \Psi(\widetilde{F}_1)$. Let $\widehat{\widetilde{F}}_1$ be a universal approximator of $\widetilde{F}_1$ and $((V_k^{(1)})^n)^n$ be the $k$-th hiddent layer of $\widehat{F} = \Psi(\widehat{\widetilde{F}}_1)$.*

*Then, the number of free parameters of the affine map from the input layer $V^n$ to the first hidden layer $((V_1^{(1)})^n)^n$ is $5 \dim V \dim V_1^{(1)} + 2 \dim V_1^{(1)}$. Moreover, the number of free parameters of the affine map from the $(k-1)$-th hidden layer $((V_{k-1}^{(1)})^n)^n$ to the $k$-th hidden layer $((V_k^{(1)})^n)^n$ for $k = 2, \ldots, d$ is at most $21 \dim V_{k-1}^{(1)} \dim V_k^{(1)} + 2 \dim V_k^{(1)}$.*

*Proof.* For $G = S_n$, $\mathrm{Stab}_G(1) \simeq S_{n-1}$ holds and the left coset decomposition is $S_n = \bigsqcup_{i=1}^n \mathrm{Stab}_G(1)(1\ i)$. Thus, the first hidden layer is isomorphic to $\mathbb{R}[S_n] \otimes_{\mathbb{R}[S_{n-1}]} (V_1^{(1)})^n$ for a finite vector space $V_1^{(1)}$.

By permutation, $S_n$ acts on the part $\mathbb{R}^n$ part of $V^n \simeq \mathbb{R}^n \otimes_{\mathbb{R}} V$. By the argument of Young tableau (c.f. (James, 1987, 9.2)), the permutation representation on $\mathbb{R}^n$ is the sum of two irreducible representations corresponding to the partitions $\lambda_1 = (n)$ (trivial representation) and $\lambda_2 = (n-1, 1)$ of $n$. On the other hand, the representation on $\mathbb{R}[S_n] \otimes_{\mathbb{R}[S_{n-1}]} (V_1^{(1)})^n \simeq \mathbb{R}[S_n] \otimes_{\mathbb{R}[S_{n-1}]} \mathbb{R}^n \otimes_{\mathbb{R}} V_1^{(1)}$ is the induced representation of the restricted representation of the permutation representation on $\mathbb{R}^n$. By the argument of Young tableau again (c.f. (James, 1987, 9.2)), the restricted representation is the sum of irreducible representations corresponding to the partitions $\lambda_{11} = (n-1)$, $\lambda_{21} = (n-1)$, and $\lambda_{22} = (n-2, 1)$ of $n-1$. Then, the induced representation of it is the sum of irreducible representations corresponding to the partitions $\lambda_{111} = (n)$, $\lambda_{112} = (n-1, 1)$, $\lambda_{211} = (n)$, $\lambda_{212} = (n-1, 1)$, $\lambda_{221} = (n-1, 1)$, $\lambda_{222} = (n-2, 2)$, and $\lambda_{223} = (n-2, 1, 1)$. Let $V_\lambda$ be the irreducible representation over $\mathbb{R}$ corresponding to the partition $\lambda$ of $n$. Then, we have the irreducible representation decomposition

$$\mathbb{R}[S_n] \otimes_{\mathbb{R}[S_{n-1}]} \mathbb{R}^n \simeq V_{(n)}^{\oplus 2} \oplus V_{(n-1,1)}^{\oplus 3} \oplus V_{(n-2,2)} \oplus V_{(n-2,1,1)}. \tag{22}$$

The permutation representation can be decomposed into

$$\mathbb{R}^n \simeq V_{(n)} \oplus V_{(n-1,1)}.$$

We set $\mathrm{Hom}_G(V, V')$ as the set of $G$-equivariant linear maps from an $\mathbb{R}[G]$-module $V$ to an $\mathbb{R}[G]$-module $V'$. Then, by Schur's lemma ((Curtis & Reiner, 1966, (27.3))), $\mathrm{Hom}_{S_n}(V_\lambda, V_{\lambda'}) = 0$ holds if $\lambda \neq \lambda'$. Moreover, it is also known that

$$\dim_{\mathbb{R}} \mathrm{Hom}_{S_n}(V_{(n)}, V_{(n)}) = \dim_{\mathbb{R}} \mathrm{Hom}_{S_n}(V_{(n-1,1)}, V_{(n-1,1)}) = 1 \tag{23}$$

holds. Indeed, it is known that

$$V_{(n)} \simeq \{c\, \mathbf{1} \in \mathbb{R}^n \mid c \in \mathbb{R}\} \text{ and}$$
$$V_{(n-1,1)} \simeq \{\boldsymbol{X} \in \mathbb{R}^n \mid \mathbf{1}^\top \boldsymbol{X} = 0\},$$

where $\mathbf{1} \in \mathbb{R}^n$ is the all one vector and $\mathbf{1}^\top$ is the transposition of vector $\mathbf{1}$ and we can show directly that $\mathrm{Hom}_{S_n}(V_{(n)}, V_{(n)}) \simeq \mathbb{R}$ and $\mathrm{Hom}_{S_n}(V_{(n-1,1)}, V_{(n-1,1)}) = \mathbb{R}$. More explicitly, we can show that

$$\mathrm{Hom}_{S_n}\left(V_{(n)} \oplus V_{(n-1,1)}, V_{(n)} \oplus V_{(n-1,1)}\right) \simeq \mathbb{R}\left(\frac{1}{n}\mathbf{1}\mathbf{1}^\top\right) \oplus \mathbb{R}\left(I - \frac{1}{n}\mathbf{1}\mathbf{1}^\top\right) \subset \mathbb{R}^{n\times n}$$

where $I \in \mathbb{R}^{n\times n}$ is the unit matrix, and $(1/n)\mathbf{1}\mathbf{1}^\top$ (resp. $I - (1/n)\mathbf{1}\mathbf{1}^\top$) is the identity on $V_{(n)}$ (resp. $V_{(n-1,1)}$) and the zero map on $V_{(n-1,1)}$ (resp. $V_{(n)}$). This implies that

$$\mathrm{Hom}_{S_n}(V^n, \mathbb{R}[S_n] \otimes_{\mathbb{R}[S_{n-1}]} (V_1^{(1)})^n)$$
$$\simeq \mathrm{Hom}_{S_n}(\mathbb{R}^n \otimes_{\mathbb{R}} V, \mathbb{R}[S_n] \otimes_{\mathbb{R}[S_{n-1}]} \mathbb{R}^n \otimes_{\mathbb{R}} V_1^{(1)})$$
$$\simeq \mathrm{Hom}_{S_n}(\mathbb{R}^n, \mathbb{R}[S_n] \otimes_{\mathbb{R}[S_{n-1}]} \mathbb{R}^n) \otimes_{\mathbb{R}} \mathrm{Hom}(V, V_1^{(1)})$$
$$\simeq (\mathrm{Hom}_{S_n}(V_{(n)}, V_{(n)}^{\oplus 2}) \oplus \mathrm{Hom}_{S_n}(V_{(n-1,1)}, V_{(n-1,1)}^{\oplus 3})) \otimes_{\mathbb{R}} \mathrm{Hom}(V, V_1^{(1)}). \tag{24}$$

Thus, by equation (23), the dimension of $\mathrm{Hom}_{S_n}(V^n, \mathbb{R}[S_n] \otimes_{\mathbb{R}[S_{n-1}]} (V_1^{(1)})^n)$ is equal to $5 \dim V \dim V_1^{(1)}$.

Next, we consider the bias vector $(\boldsymbol{B}_j)_{j=1}^n \in ((V_1^{(1)})^n)^n \simeq \mathbb{R}[S_n] \otimes_{\mathbb{R}[S_{n-1}]} (V_1^{(1)})^n$, where $\boldsymbol{B}_j = (\boldsymbol{b}_{j1}, \ldots, \boldsymbol{b}_{jn}) \in (V_1^{(1)})^n$. Then, by the action $*$ of $S_n$, $(\boldsymbol{B}_j)_{j=1}^n$ is invariant, i.e.,

$$\sigma * (\boldsymbol{B}_j)_{j=1}^n = ((\sigma_{1j}\sigma\sigma_{1\phi_\sigma^{(1)}(j)}^{-1}) \cdot \boldsymbol{B}_{\phi_\sigma^{(1)}(j)})_{j=1}^n = (\boldsymbol{B}_j)_{j=1}^n.$$

holds for any $\sigma \in S_n$.

Without of loss of generality, we may assume that $\sigma_{11}$ is the unit element $e$. We consider $\sigma = \sigma_{1j_0}^{-1}\tau\sigma_{11}$ for $\tau \in H_1 = \mathrm{Stab}_{S_n}(1)$, we have $\sigma_{1j_0}\sigma = \tau\sigma_{11} = \tau$. In particular, $\phi_\sigma^{(1)}(j_0) = 1$ Thus, $j_0$-th entry of $(\sigma_{1j}^{-1}\tau) * (\boldsymbol{B}_j)_{j=1}^n$ becomes

$$(\sigma_{1j_0}\sigma\sigma_{1\phi_\sigma^{(1)}(j_0)}^{-1}) \cdot \boldsymbol{B}_{\phi_\sigma^{(1)}(j_0)} = \tau \cdot \boldsymbol{B}_1. \tag{25}$$

Because $(\boldsymbol{B}_j)_{j=1}^n$ is invariant by the action $*$ of $S_n$,

$$\boldsymbol{B}_{j_0} = \tau \cdot \boldsymbol{B}_1$$

holds for any $j_0 = 1, 2, \ldots, n$ and $\tau \in \mathrm{Stab}_{S_n}(1)$. This implies that by $\tau = e$, $\boldsymbol{B}_j = \boldsymbol{B}_1$ holds for any $j = 1, 2, \ldots, n$. Furthermore, $\tau \cdot \boldsymbol{B}_1 = \boldsymbol{B}_1$ holds for any $\tau \in \mathrm{Stab}_{S_n}(1)$. Because the orbit decomposition of $\{1, 2, \ldots, n\}$ by $H_1 = \mathrm{Stab}_{S_n}(1)$ is

$$\{1, 2, \ldots, n\} = \{1\} \sqcup \{2, 3, \ldots, n\},$$

$\boldsymbol{B}_1 = (\boldsymbol{b}_{11}, \boldsymbol{b}_{12}, \ldots, \boldsymbol{b}_{1n})$ satisfies that $\boldsymbol{b}_{12} = \cdots = \boldsymbol{b}_{1n}$. This means that the number of free parameters of $\boldsymbol{B}_1$ (hence, of $(\boldsymbol{B}_j)_{j=1}^n$) is $2 \dim V_1^{(1)}$.

Thus, the number of free parameters of the affine map from the input layer to the first hidden layer is $5 \dim V \dim V_1^{(1)} + 2 \dim V_1^{(1)}$.

Next, we consider the linear part of the affine map from the $(k-1)$-th layer $((V_{k-1}^{(1)})^n)^n$ to the $k$-th layer $((V_k^{(1)})^n)^n$ for $k > 1$. Then, by Theorem 5, $S_n$ acts on these layers by the action $*$, and the representations defined by this action $*$ is isomorphic to the sum of the induced representation of the restricted representation. by equation (22) and a similar argument as equation (24), we have

$$\mathrm{Hom}_{S_n}(\mathbb{R}[S_n] \otimes_{\mathbb{R}[S_{n-1}]} (V_{k-1}^{(1)})^n, \mathbb{R}[S_n] \otimes_{\mathbb{R}[S_{n-1}]} (V_k^{(1)})^n)$$
$$\simeq \mathrm{Hom}_{S_n}(\mathbb{R}[S_n] \otimes_{\mathbb{R}[S_{n-1}]} \mathbb{R}^n \otimes_{\mathbb{R}} V_{k-1}^{(1)}, \mathbb{R}[S_n] \otimes_{\mathbb{R}[S_{n-1}]} \mathbb{R}^n \otimes_{\mathbb{R}} V_k^{(1)})$$
$$\simeq \mathrm{Hom}_{S_n}(\mathbb{R}[S_n] \otimes_{\mathbb{R}[S_{n-1}]} \mathbb{R}^n, \mathbb{R}[S_n] \otimes_{\mathbb{R}[S_{n-1}]} \mathbb{R}^n) \otimes_{\mathbb{R}} \mathrm{Hom}(V_{k-1}^{(1)}, V_k^{(1)})$$
$$\simeq (\mathrm{Hom}_{S_n}(V_{(n)}^{\oplus 2}, V_{(n)}^{\oplus 2}) \oplus \mathrm{Hom}_{S_n}(V_{(n-1,1)}^{\oplus 3}, V_{(n-1,1)}^{\oplus 3})$$
$$\oplus \mathrm{Hom}_{S_n}(V_{(n-2,2)}, V_{(n-2,2)}) \oplus \mathrm{Hom}_{S_n}(V_{(n-2,1,1)}, V_{(n-2,1,1)})) \otimes_{\mathbb{R}} \mathrm{Hom}(V_{k-1}^{(1)}, V_k^{(1)}).$$

Because the dimension of the division algebra over $\mathbb{R}$ is at most 4, the dimensions of $\mathrm{Hom}_{S_n}(V_{(n-2,2)}, V_{(n-2,2)})$ and $\mathrm{Hom}_{S_n}(V_{(n-2,1,1)}, V_{(n-2,1,1)}))$ are at most 4. Thus, the dimension of $\mathrm{Hom}_{S_n}(((V_{k-1}^{(1)})^n)^n, ((V_k^{(1)})^n)^n)$ is at most $21 \dim V_{k-1}^{(1)} \dim V_k^{(1)}$.

On the other hand, the number of free parameters of the bias vectors is $2 \dim V_k^{(1)}$ by the similar argument as above. $\qquad \square$

