# OpenReview forum: "Decomposition of Equivariant Maps via Invariant Maps: Application to Universal Approximation under Symmetry."
_TMLR — Accepted by TMLR_

### Review · Reviewer_NyZF · 2024-04-22

**Summary Of Contributions:**

Given a fixed group G acting on a function space, the paper studies the problem of representing G-equivariant and related invariant maps using deep neural networks.
The main technical result of the paper is a correspondence between G-equivariant maps and H-invariant maps, where H is a stabilizer subgroup of G. Using this correspondence, the authors are able to provide new universal approximation theorems for equivariant maps, when a corresponding theorem is known for invariant maps with respect to stabilizer subgroups.

**Audience:**

Yes

**Claims And Evidence:**

Yes

**Requested Changes:**

I have several concerns about the novelty and the contributions of the paper and I would be happy to hear the authors' response on these matters:

1. First, if a group $G$ acts on a space $Y$ (say transitively) and $H$ is the stabilizer subgroup of $G$, one can think about the question of relating $G$-equivariant maps to $H$-invariant maps, as evoking a relationship between the representations of $G$ and the representations of $H$. Connecting between representations of a group and its subgroup, through inductions and restrictions, is a very old question in representation theory which goes back to the Frobenius reciprocity theorem.
Given that, I suspect that the correspondence the authors claim as their main result is a known result in the representation theory community. I have searched the literature for this result and while I could not find the exact statement, I did find some similar results. In fact, there is a way to realize the main result of the paper as a special case of the results I found in the literature.
For example, Theorem 2.3.5 (which states and proves the result, but is certainly not the origin of the theorem) from the following thesis https://phaidra.univie.ac.at/detail/o:1299980, is a much more general statement about the relationship between invariant and equivariant maps. In the case where $G$ acts transitively on $Y$, we can replace $X$ by $V^X$ in Theorem 2.3.5 and take $Y = G/H$ with $W = E_0$. In this case, the space of section $\Gamma(E)$ is precisely functions from Y to W, and we recover the Theorem 1. To cover the general case, when the action of $G$ is not transitive, we can now apply the theorem to each orbit separately.
Let me point out that the fact that Theorem 1 is not new does not mean it is not useful. I do think that there is some merit in the exact statement of Theorem 1 which is useful for problems more specific to learning theory. I also think that there is some benefit in presenting and explaining the result to the learning community, which, in general, is not well-initiated in representation theory. However, what the paper misses at the moment is a **serious** discussion about the context, history, and related literature of the result.

2.  I feel that some of the discussion around Theorem 2 can sometimes be misleading. Rather than being a **universal** approximation result it is a **conditional** approximation result. The authors indeed establish that equivariant maps can always be approximated by the corresponding invariant maps (I'm not sure I would call this universal). However, in order to get an approximation result by some hypothesis class, like neural networks, one must first prove a universal approximation result for invariant maps (resp. equivariant maps) for the given class, and then use the Theorem to move to equivariant maps (resp. invariance maps). It seems to me that my last point should be emphasized better and discussed. I don't think it is true that there is a general universal approximation theorem for equivariant maps of **general** group actions.

3. In line with my last comment, the applications the authors present are all based on existing approximation results. Due to this and because of the current writing it is sometimes hard to decipher which results are new and which results were already known. A more in-depth comparison with the literature is in order. In particular, it is important to explain which results from Sections 4 and 5 already appear in the literature, perhaps with different architectures, both from a quantitative and a qualitative perspective.

**Strengths And Weaknesses:**

Strengths:
- The topic of learning and representing functions under group symmetry has gained considerable traction recently and is of interest across the ML community.
- The current work proposes a somewhat general principle that connects invariant and equivariant maps, potentially addressing a wide range of cases.

Weakness:
- My primary concern revolves around the novelty of the work, which I will elaborate on in the next section.

---

> ### Author Response · Authors · 2024-07-17
> **Response to reviewer NyZF**
>
> We thank Reviewer NIF for the thorough review and valuable suggestions to improve the paper. We would like to respond to the 3 requested changes.
>
> > 1. First, if a group 𝐺 acts on a space 𝑌 (say transitively) and 𝐻 is the stabilizer subgroup of 𝐺, one can think about the question of relating 𝐺-equivariant maps to 𝐻-invariant maps, as evoking a relationship between the representations of 𝐺 and the representations of 𝐻. Connecting between representations of a group and its subgroup, through inductions and restrictions, is a very old question in representation theory which goes back to the Frobenius reciprocity theorem. Given that, I suspect that the correspondence the authors claim as their main result is a known result in the representation theory community. I have searched the literature for this result and while I could not find the exact statement, I did find some similar results. In fact, there is a way to realize the main result of the paper as a special case of the results I found in the literature. For example, Theorem 2.3.5 (which states and proves the result, but is certainly not the origin of the theorem) from the following thesis [https://phaidra.univie.ac.at/detail/o:1299980](https://phaidra.univie.ac.at/detail/o:1299980), is a much more general statement about the relationship between invariant and equivariant maps. In the case where 𝐺 acts transitively on 𝑌, we can replace 𝑋 by 𝑉𝑋 in Theorem 2.3.5 and take 𝑌=𝐺/𝐻 with 𝑊=𝐸0. In this case, the space of section Γ(𝐸) is precisely functions from Y to W, and we recover the Theorem 1. To cover the general case, when the action of 𝐺 is not transitive, we can now apply the theorem to each orbit separately. Let me point out that the fact that Theorem 1 is not new does not mean it is not useful. I do think that there is some merit in the exact statement of Theorem 1 which is useful for problems more specific to learning theory. I also think that there is some benefit in presenting and explaining the result to the learning community, which, in general, is not well-initiated in representation theory. However, what the paper misses at the moment is a **serious** discussion about the context, history, and related literature of the result.
>
> After reviewing the mathematical content, we agree with the assertion from the Reviewer. It appears that the result we proved is known in the context of homogeneous vector bundles as a "geometric Frobenius reciprocity law". Indeed this variant of the Frobenius reciprocity law is stated in Theorem 1.1.4 of [1] as a consequence of the homogeneous vector bundle version of our statement. Thus, we think that adding a paragraph discussing the above connection is necessary for the paper, and we intend to do so in a revised version.
>
> Yet, as mentioned by the reviewer, we think that our formulation and proof of the result in elementary words are valuable for the audience to whom this article is addressed. They require minimal technical background, and thus, are more enlightening.<br>
> We added a paragraph discussing this below the proof of the theorem.
>
> > 2. I feel that some of the discussion around Theorem 2 can sometimes be misleading. Rather than being a **universal** approximation result it is a **conditional** approximation result. The authors indeed establish that equivariant maps can always be approximated by the corresponding invariant maps (I'm not sure I would call this universal). However, in order to get an approximation result by some hypothesis class, like neural networks, one must first prove a universal approximation result for invariant maps (resp. equivariant maps) for the given class, and then use the Theorem to move to equivariant maps (resp. invariance maps). It seems to me that my last point should be emphasized better and discussed. I don't think it is true that there is a general universal approximation theorem for equivariant maps of **general** group actions.
>
> The remark of the reviewer is accurate. Our result about approximation of equivariant maps only holds provided that universal approximation by invariant maps has been previously established, and vice versa. However, we claim that this condition is always fulfilled, perhaps by architectures that are only theoretical and not implementable. For example, it suffices to consider universal classical multi-layer perceptrons, and to do an averaging over the full group, or to use a model based on Reynolds operator [2,3]. For that reason, we may talk about universal approximation for equivariant maps.
>
> To clarify this point, we added a paragraph after the proof of the theorem.

---

> ### Author Response · Authors · 2024-07-17
>
> > 3. In line with my last comment, the applications the authors present are all based on existing approximation results. Due to this and because of the current writing it is sometimes hard to decipher which results are new and which results were already known. A more in-depth comparison with the literature is in order. In particular, it is important to explain which results from Sections 4 and 5 already appear in the literature, perhaps with different architectures, both from a quantitative and a qualitative perspective.
>
> In section 4, we do not pretend to prove new results. We simply consider these as straightforward examples in an attempt to highlight the usefulness of our theorems.
>
> In section 5, we indeed extend a result from [5]. The approximation rate from 5 holds for invariant networks. We prove an extension to equivariant networks thanks to our theory.
>
>
>
> # References
> [1] Parabolic Geometries I: Background and General Theory, Andreas Cap and Jan Slovak, AMS mathematical surveys and monographs 154, 2009.
> [2] Frame averaging for invariant and equivariant neural networks, Puny et. al., ICLR 2022.
> [3] Invariant and equivariant Reynolds networks, Sannai et. al., JMLR, 2024.
> [4] Universal invariant and equivariant graph neural networks, Keriven et. al. NeurIPS 2019.
> [5] Improved generalization bounds of group invariant/equivariant deep networks via quotient feature spaces. Akiyoshi Sannai, Masaaki Imaizumi, and Makoto Kawano.Uncertainty in Artificial Intelligence, pp.771–780. PMLR, 2021

---

### Review · Reviewer_LapW · 2024-05-28

**Summary Of Contributions:**

This article studies how to construct equivariant maps through invariant maps with regard to a group. Based on the sub-group structure of a group, the relationship between these maps are established. From this, universal approximation of equivariant maps are obtained based on existing results for invariant maps. Further discussion on the complexity of the universal approximation are analyzed for finite group.

**Audience:**

Yes

**Claims And Evidence:**

Yes

**Requested Changes:**

Define what is Stab_G(y_i) in Theorem 1 before-hand.
Typo: know -> known on page 7.
Is there any reduction of number of parameter in the case G= translation group, compared to fully connected neural networks ?
Clarify in Remark 1 or latter the relation to convolutional neural networks.

**Strengths And Weaknesses:**

Strength: This article is well written, and the obtained main result provides a novel way to build equivariant maps. Even though the results are theoretical, the results are made easy to understand through illustrative examples.

Limitation: There is one case which is important to discuss, in relation to convolutional neural networks. I am wondering if you take G = translation group on X=Y=R^2, what will happen to Theorem 1? Is Stab_G(0) = Id, so that I do not see the connection of the proposed way to construction translation equivariant maps to the convolutional layer in convolutional neural networks. Therefore the theory seems limited to include known architectures.

---

> ### Author Response · Authors · 2024-07-17
> **Response to reviewer LapW**
>
> We thank Reviewer LapW for the valuable feedback and suggestions to improve the paper.
> We would like to respond to the questions and concerns raised by the reviewer.
>
> ### Limitation
>
> > There is one case which is important to discuss, in relation to convolutional neural networks. I am wondering if you take G = translation group on X=Y=R^2, what will happen to Theorem 1? Is Stab_G(0) = Id, so that I do not see the connection of the proposed way to construction translation equivariant maps to the convolutional layer in convolutional neural networks. Therefore the theory seems limited to include known architectures.
>
> This is an interesting point.
> Convolutional layer is made of convolution product, non linearity and pooling. Let's just consider the linear part of the convolutional layer, which is the convolution product. The question is: can we recover the convolution product against a kernel from our theorem?
>
> Let $X=Y=\mathbb{R^2}$, $W=V=\mathbb{R}$, and say an image is an element of $\mathbb{R}^{\mathbb{R^2}}$.
> The translation group $\mathbb{T}$ is considered to be $G=\mathbb{T}=\{ T_{\bf{v}} | \bf{v} \in\mathbb{R}^2\}$. In this case, surely $\text{Stab}_G(x)=\{\text{Id}\}$ holds for any $x$ and the orbit $Gx=\mathbb{R}^2$.
>
> If we restrict to linear transformation, then our theorem says that an equivariant linear map $F \in \mathscr{L}\left(\mathbb{R}^{\mathbb{R}^2} ,\mathbb{R}^{\mathbb{R}^2}\right)$ is bijectively mapped to a $\{\mathrm{Id} \}$-invariant map $h \in\mathscr{L}\left(\mathbb{R}^{\mathbb{R}^2} ,\mathbb{R}\right)$.
> But $\{\mathrm{Id} \}$-invariant maps are just ordinary maps and $\mathscr{L}\left(\mathbb{R}^{\mathbb{R}^2} ,\mathbb{R}\right)$ is isomorphic to $\mathbb{R}^{\mathbb{R}^2}$, so $h$ is just an image.
>
> Does this $h$ correspond to the kernel against which $F$ is a convolution product, i.e. $F(f)= f \star h:x \mapsto \int f(t) h(x-t)dt$ ? We don't know and we could not turn this heuristic into a rigorous proof. We leave it to future work.
>
> ### Requested changes
>
> > - Define what is Stab_G(y_i) in Theorem 1 before-hand.
>
> We added a sentence to define the stabilizers before stating the theorem.
>
> > - Is there any reduction of number of parameter in the case G= translation group, compared to fully connected neural networks?Clarify in Remark 1 or latter the relation to convolutional neural networks.
>
> If neural network models with the action by the translation group $\mathbb{T}$ could be constructed (even by discrete approximation), the number of parameters should be reduced less than fully connected models, at least as discussed here. However, we have not been able to formulate it precisely, so we did not go into it deeply this time. This is also a subject for future research.

---

### Review · Reviewer_3BTQ · 2024-07-03

**Summary Of Contributions:**

The authors theoretically study the relationship between equivariant and invariant learning tasks and show a different approach to obtaining a universal approximation theorem under these symmetries. More precisely,

1. They show that any G-equivariant task can be reduced to some Stabiliser(G)-invariant task, for any group G.
2. Moreover, using this they show universal approximation theorems under symmetry for finite groups, going beyond just $S_n$ (cf.Sannai et al.2021) for both invariant tasks and equivariant tasks.
3. They particularly rely on their reduction to the invariant tasks to argue about the universal approximation theorem for equivariant deep networks, which is a complementary approach to methods used in works such as (Yarotsky 2022).

**Audience:**

Yes

**Broader Impact Concerns:**

None.

**Claims And Evidence:**

Yes

**Requested Changes:**

Questions:
1. Could the authors explain any added benefit of this reduction and the applications to the universal approximations, apart from the existing approaches for the equivariant tasks being complicated. It could help if they mention how their approach helps alleviate these complications?

2. Do the authors think if the universal approximators are merely a theoretical construct, or such architectures which have fewer parameters than fully connected networks be trained for such equivariant/invariant tasks?

3. I think the theoretical explanation of Remark 1 provided in the appendix, while it is useful does not generally help improve the intuition about the remark. Maybe having some pictorial descriptions of the ``differences" in the architectures from previous work and this paper would help significantly.

4. As an extension of the previous point, I think this paper is not an easy read for those without any abstract algebra background. They have a 2 page introduction to Groups and orbits, but they dont seem to suffice. In Section B, in the appendix, they start talking about modules and automorphisms, the definitions of which are not necessarily contained in this paper. It would be great if the authors could consider adding some more basic definitions and properties and keep the paper as self-contained as possible.

5. The applications targeting universal approximation results are described for finite groups. It would be nice to understand the challenges to extend this for infinite groups/particular interesting infinite groups, such as rotations etc.

Minor comments/typos:

1. Page 3, first paragraph: which was mention as a possible... -> mentioned
2. Page 3, first paragraph: with a plethora of various architecture... -> architectures
3. It would help to add a bit more intuition about why equation (1) is defined as is.
4. In the orbit notation in page 18, it is better to remove the $\sigma.x$ in between.
5. In Section 3.1, first line, just before the equation, there is an additional white space before the words "equation 1".
6. In page 19, right after the Example 6, small $h$ is defined, but never used and in that sentence it is not clear what is $\tau$ and what is $h$, the equation following it doesnt seem to make sense.
7. In page 20, right after equation 16, the word equation appears twice.

**Strengths And Weaknesses:**

Strengths:

The main strength is showing the relation between equivariant and invariant tasks as stated in their main theorem, using various algebraic techniques. Then they use it to provide universal approximation theorems for architectures with such symmetries, which allows them to argue about equivariant tasks through invariant tasks (for which universal approximation theorems are known for important finite groups such as $S_n$).

Weaknesses:

1. While their Theorem 1 talks about the reduction from equivariant to certain invariant tasks, their universal approximation results only seem to hold for finite groups.
2. In Remark 1, they mention that their proofs talk about certain equivariant deep network architectures that are different from the existing architectures and that the parameters are less than that of those existing architectures described in (Zaheer et al. 2017) and (Maron et al. 2019).

---

> ### Author Response · Authors · 2024-07-17
> **Response to reviewer 3BTQ**
>
> We thank Reviewer 3BTQ for the valuable feedback and suggestions to improve the paper.
> We would like to respond to the questions raised by the reviewer.
>
> ### Weaknesses
> > 2. In Remark 1, they mention that their proofs talk about certain equivariant deep network architectures that are different from the existing architectures and that the parameters are less than that of those existing architectures described in (Zaheer et al. 2017) and (Maron et al. 2019).
>
> It is not totally clear to us what is the weakness the reviewer is pointing out here. Perhaps, there is a misunderstanding. In terms of the number of parameters, we do not compare our equivariant model to those of Zaheer et al. 2017 or Maron et al. 2019. We compare it to a fully connected neural network. When incorporating symmetries, we expect the number of parameters to be reduced compared to a fully connected layer of similar width, and this is the fact we verify in this part of the paper.
> In the revised version, we rewrote the Remark 1 in an attempt to make it more clear to the reader.
>
> ### Questions
> > 1. Could the authors explain any added benefit of this reduction and the applications to the universal approximations, apart from the existing approaches for the equivariant tasks being complicated. It could help if they mention how their approach helps alleviate these complications?
>
> The main benefit of this reduction is to enable transferring the study of equivariant problems to the study of invariant problems and vice versa. Throughout, the paper, we insist on the application of this principle to prove universal approximation results. There are other applications that we have not explored here. For instance, training: one could use the reduction to train an invariant model, and then turn it into an equivariant via our construction.
>
> Apart from the equivariance-invariance relation, we have other possible benefit in mind.
>
> One of the characteristics of the model in this paper is that the action of the group in the hidden layers is different from the original one, as described in Remark 1. As a result, it can be seen as relaxing some of the constraints of the conventional equivariant models in which the action is the same in all layers. We believe that what matters for an equivariant layer is the commutativity between the action and the affine map, even if the actions are different between the input and the output spaces.
>
> Another aspect of the Theorem is to link symmetries of the full group to those of some subgroups. Suppose a scenario when one does not know the full group of symmetries of a problem, but can only assume to know some subgroups. Our decomposition theorem could help recover the full group of equivariance from the knowledge of some subgroups' invariance.
>
> >2. Do the authors think if the universal approximators are merely a theoretical construct, or such architectures which have fewer parameters than fully connected networks be trained for such equivariant/invariant tasks?
>
> Although universal approximators are basically a theoretical construct, any model created by the construction method proposed here is a model equipping with equivariance a priori. Therefore, it may be easier to naturally fit a fully connected neural network model to an equivariant task than to acquire robustness to group actions by training a fully connected neural network model to an equivariant task through data augmentation, etc., as has been done in the past.
>
> Furthermore, as we suggested in the response to the previous question, using the architecture in this paper, if learned architectures with invariance have been obtained, it is possible to construct an equivariant model by plugging them in. Although there are some interesting questions such as an experimental investigation of the practicality of the proposed model and a demonstration of whether the weight matrix of the learned neural networks model naturally becomes similar to the architecture described in this paper, we would like to leave them as future works.
>
> > 3. I think the theoretical explanation of Remark 1 provided in the appendix, while it is useful does not generally help improve the intuition about the remark. Maybe having some pictorial descriptions of the "differences" in the architectures from previous work and this paper would help significantly.
>
> We added two commutative diagrams Remark 1. We hope that these diagrams will be helpful to understand intuitively.

---

> > ### Author Response · Authors · 2024-07-17
> >
> > > 4. As an extension of the previous point, I think this paper is not an easy read for those without any abstract algebra background. They have a 2 page introduction to Groups and orbits, but they don't seem to suffice. In Section B, in the appendix, they start talking about **modules** and **automorphisms**, the definitions of which are not necessarily contained in this paper. It would be great if the authors could consider adding some more basic definitions and properties and keep the paper as self-contained as possible.
> >
> > We added some definitions of words in this paper (e.g., modules, automorphisms, isomorphic, intertwining operators, and so on) to Appendix B. Moreover, we rewrote the explanation so that there is no leap of argument.
> >
> > > 5. The applications targeting universal approximation results are described for finite groups. It would be nice to understand the challenges to extend this for infinite groups/particular interesting infinite groups, such as rotations etc.
> >
> > As mentioned in the reply to the 2nd Reviewer, if neural network models with the action by the translation group $\mathbb{T}$ and $\text{SO}_2(\mathbb{R})$ could be constructed (even by discrete approximation), the number of parameters should be reduced less than fully-connected models, at least as discussed here. However, we have not been able to formulate it precisely, so we did not go into it deeply this time. This is also a subject for future research.
> >
> > ### Minor comments
> >
> > > 3. It would help to add a bit more intuition about why equation (1) is defined as is.
> >
> > We assume that the reviewer refers to the fact that the action in equation (1) is defined via the inverse $\sigma^{-1}$ rather than $\sigma$ itself. We added an explanation about this in the form of a footnote.

---

### Author Response · Authors · 2024-07-17
**Revised version**

We thank the reviewers for their reviews and for spotting the typos.

We have revised the paper accordingly. The substantial changes appear in blue.

---

### Decision · Action_Editor_9K6C · 2024-09-12

**Recommendation:** Accept as is

**Comment:**

The paper studies links between equivariant and invariant architectures, and applies this to obtain approximation results for equivariant maps whenever universal approximation holds for corresponding invariant maps (and vice versa). The reviewers found the theoretical results and their implications interesting and relevant for the community, and were satisfied with the authors' responses. I recommend acceptance.

**Audience:**

Yes.

**Claims And Evidence:**

This is a theory paper about neural network approximation, and the main results are supported by appropriate proofs.